# Dynamics of Research into Modeling the Power Consumption of Virtual Entities Used in the Telco Cloud

**DOI:** 10.3390/s23010255

**Published:** 2022-12-26

**Authors:** Etienne-Victor Depasquale, Franco Davoli, Humaira Rajput

**Affiliations:** 1Department of Communications and Computer Engineering, University of Malta, MSD 2080 Msida, Malta; 2The National Laboratory of Smart and Secure Networks (S2N), Italian National Consortium for Telecommunications (CNIT), Department of Electrical, Electronic and Telecommunications Engineering and Naval Architecture (DITEN), University of Genoa, 16145 Genoa, Italy; 3Department of Electrical, Electronic and Telecommunications Engineering and Naval Architecture (DITEN), University of Genoa, 16145 Genoa, Italy

**Keywords:** virtualization, power consumption, power models, power meters, energy-aware algorithms

## Abstract

This article is a graphical, analytical survey of the literature, over the period 2010–2020, on the measurement of power consumption and relevant power models of virtual entities as they apply to the telco cloud. We present a novel review method, that summarizes the ***dynamics*** as well as the results of the research. Our method lends insight into trends, research gaps, fallacies and pitfalls. Notably, we identify limitations of the widely used linear models and the progression towards Artificial Intelligence/Machine Learning techniques as a means of dealing with the seven major dimensions of variability: workload type; computer virtualization agents; system architecture and resources; concurrent, co-hosted virtualized entities; approaches towards the attribution of power consumption to virtual entities; frequency; and temperature.

## 1. Introduction

Several surveys of the results (see Section 1.6, “Related Surveys”) of research into modeling the power consumption of virtual entities (VEs, i.e., virtual machines (VMs) or containers) have been written. In this work, our contribution lies in ***a thorough analysis of the dynamics of research itself: the challenges, the approaches, the pitfalls, the fallacies, and the research gaps, without neglecting the fruits of the research***. Our intended audience is the prospective researcher, seeking to understand the dynamics of research into the predictive modeling and supporting measurements of power consumption by individual VEs relevant to ***the telco cloud***. Dynamics are characterized through a thorough frequency analysis, which we conduct **through the application of a novel method we have developed** [1] that is unique in its ability to ***parse*** research literature. Through the visual aids we provide, and our observations through cross-cutting themes, a prospective researcher obtains a thorough characterization of the problems, approaches, developments, formal methods, pitfalls, fallacies and research gaps that characterize this research space.

**Among the themes** that our survey has brought us to identify, we have pointed out that all the problem categories we identified touch one or more of a set of seven main variables that may affect power consumption by virtual entities and the ensuing model representations: workload type, characteristics of the virtualization agent (VM or container), host machine resources and architecture, temperature, operating frequency, attribution of a fraction of consumed power to individual VEs and the mutual influence of concurrent VEs.

**Among the major pitfalls** that emerged from our thematic analysis, we highlight here the misconception of the Data Plane Development Kit’s (DPDK) power efficiency (commonly misportrayed as a power hog), the often-unacknowledged limitations of the widely used linear models, the problematic use of benchmarks in model validation, the failure to precisely identify the physical contexts of some experimental research, the influence of synthetic workload generators on measurements and the sometimes-overlooked relevance of processor organization on power consumption measurements. We have also pointed out the unavoidable need to precisely identify the scope and limitations of models and the fallacy of the quest for a “universal” power model.

**The research gaps** we identify are four: (a) modeling of containers’ power consumption; (b) the effect of overcommitment on power efficiency; (c) investigation and classification of DPDK applications; and (d) modeling of power consumption by virtualized I/O (a challenge which is starting to receive some attention).

### 1.1. Why Is Research Needed?

Precise measurement of a VE’s power consumption is difficult since measurements of its host’s power consumption cannot be related directly to it. Hardware power meters are incapable of measuring the power consumption of individual VEs co-hosted on a physical machine. Moreover, power consumption of a VE varies with its hosting machine. Therefore, for VEs, accurate measurement is predicated upon precise ***modeling*** of energy- and/or power consumption. 

A general prerequisite to devising energy- and/or power-efficient ***operations*** is accuracy in power and energy ***measurements***. With specific regard to VEs, it is also essential for billing in multi-tenant environments, so that the Infrastructure Provider (IPr) can charge customers the fair amount for the resources (including energy) they consume. Before proceeding to the scope of our survey, we pose three salient questions that frame our work.

### 1.2. Why Should Energy Consumption Be a Topic of Interest?

Energy efficiency in the Internet (and in computing and telecommunication networks in general) has become a significant problem, which has received increasing attention since the early years around 2000 (see, e.g., [2,3,4] and references therein), starting from cloud computing infrastructures, and then extending to mobile and fixed networks. Indeed, it has been shown that the smaller data centers, within which telecommunications points of presence (PoPs) may be classified, represent around 95% of the United States’ data center energy use [5]. Furthermore, this use is comparatively inefficient when compared with that of the hyperscale server farms (the remaining 5%).

### 1.3. What Is the Underlying Cause of Increased Energy Consumption?

Traffic growth is the primary cause of increased energy consumption. Table 1 shows the consistency with which Cisco’s Visual Networking Index (VNI) has been predicting heavy growth in traffic exchanged over the access network by both businesses and consumers with:endpoints over managed networks;endpoints over unmanaged networks (“Internet traffic”).

(Note that the figures refer to compound annual growth rate (CAGR); they do ***not*** refer to the percentage share of total traffic.).

The key observation lies in the realization that, year after year, significant (heavy, in the case of mobile data) growth is persistently predicted. This observation is corroborated by several other researchers, with perspectives varying from traffic at the access segment to traffic in transit between Internet Service Providers (ISPs) [10,11,12]. Cisco [6] and Sandvine [13,14] identify “video traffic” and “real-time entertainment” as the drivers of this growth. A later edition of Sandvine’s Global Internet Phenomena report [15] dedicates its executive summary exclusively to video traffic; the report shows video as consuming 60% of downstream traffic—a further 2% increase over 2018. Alcatel-Lucent Bell Labs [11] observed that growth in the metro-core due to video traffic exceeds growth in video traffic crossing the long-haul core. Clearly, it is widely recognized that video is the prime driver of this growth in traffic and attendant energy consumption and research has been addressing this problem [16,17,18,19,20,21,22,23,24,25,26,27,28].

### 1.4. How Is Energy Consumption Being Tackled?

From these (referenced) works, the development of cache architectures emerges as an important approach to controlling energy consumption, but other mechanisms exist and yet more are emerging within the general thrust towards “future networks” (ITU-T Y.3001 [29]). Networks are evolving into flexible and programmable “softwarized” virtualized infrastructures, and strongly integrated paradigms, such as 5G [30,31], rewrite the research agenda. We may thus distinguish between radical and reformist approaches:Reformist approaches seek to improve the caching of content and are characterized by their investigation of the reduction of the length of the path between the source and destination of IP traffic.Radical approaches employ the dynamic and reactive control afforded by the softwarized, virtualized infrastructures. Radical methods are enabled by standardized architectures (e.g., [32,33]) that equip the control plane with uniform interfaces that exploit extant and emerging green capabilities.

Indeed, ***the impact of virtualization technologies on power consumption in public telecommunication networks (PTNs) is still unclear***. There is a general belief that Network Functions Virtualization (NFV) should result in reduced energy consumption, owing to a consolidation of resources and increased flexibility in turning unused hardware (HW) on and off as needed. However, it is also true that “the massive introduction of general-purpose HW enabled by NFV would tend to increase power requests with respect to specialized HW solutions” [30]. Therefore, there is a need to operate power-aware management and control mechanisms in these environments. At the same time, it is necessary to limit the complexity of these mechanisms and the level of human intervention therein, to keep Operational Expenditures (OPEX) within reasonable limits. One approach to understanding this impact consists of comparative analyses of the implementations of infrastructure, with and without virtualization. This approach is taken in [34], where the evolved packet core (EPC) is studied. This work shows that the virtualized implementation is indeed less energy efficient. Unfortunately, the scope of virtualization and containerization within the converged wireless and wireline infrastructure is very broad and consideration of a single “use-case” [35] cannot be generalized to an overall statement. We therefore note that the scope for our survey needs an operational context which we suggest in the following.

### 1.5. The Scope of This Survey

#### 1.5.1. Telco Cloud: The Operations Context

We suggest that the telco cloud is our network-operations context. “Telco cloud” is an evolving notion that evokes a number of common terms in attempts to describe it. Virtualization, software-defined networking (SDN), automation and orchestration are four such terms. Other prominent terms are edge computing, containerization, microservices and resilient infrastructure [36]. We suggest three key observations that organize these terms into a coherent image of the telco cloud.

1.
**The telco cloud is, fundamentally, a hybrid cloud:**
a.Self-sourced virtualization and containerization;b.Out-sourced (public cloud) containerization.


The complementary collaboration of the PTN operator’s (PTNO) network, computing and storage infrastructure, with that of global providers of infrastructure and applications, is manifested well in [37]. A distributed cloud infrastructure operates at network (transport and interconnect) junctions. It includes (cloud) infrastructure owned and operated by the PTNO, by public cloud providers and by enterprises which consume their joint service. 

2.**The telco cloud serves both internal and external clients** [36]:
a.Internal use can suggestively be referred to as the ***IT Cloud*** [38]. This consists of applications specific to PTNOs: operational support systems (OSS) and business support systems (BSS), as well as more general applications, such as customer relationship management (CRM);b.External uses are growing organically on the basis of use cases seeded by ETSI [35] and the 5GPPP [39].3.**The service-based architecture (SBA) of the 5G Core is a good fit with cloud-native computing**. Containerization is distinctively central to cloud-native computing [40,41]. The Cloud Native Computing Foundation explicitly identifies containers as components of the approach to the concept of cloud native computing [42]. There is a clear drive towards the use of containers in lieu of virtual machines as the operating environment for network functions [40], and the 5G Core’s SBA provides a clear scope for employing containers.

We end this subsection by indicating the real estate where the VEs in our context may be deployed:**Datacenters**: here, the real estate referred to consists of points of presence (PoPs) such as metro-core PoPs at the near edge;**Softwarized and virtualized networks**: here, we refer to points of presence such as central offices (COs) and sites even deeper into the edge, such as remote radio head (RRH) sites and roadside cabinets.

#### 1.5.2. Identifying the Models in the Scope of This Study

We survey **predictive** energy and power **models**, as well as **measurements** that facilitate the qualitative and/or quantitative prediction, of consumption by individual VEs relevant to the telco cloud. A simple interpretation of the rationale that drove our selection is that we have sought works that **measure real-time power consumption by VEs** and/or model **real-time power consumption by VEs**. The object of measurements and modeling is strictly the VE.

Nonetheless, the devil is in the details and therefore the details of this simple rationale must be worked out. One important, finer point regards the VEs themselves. There are software technologies, which we shall elaborate upon in later sections, which are **functionally critical** to VEs. Works that measure, and/or model, such technologies’ power consumption are in scope. A justification of this claim on scope is not hard. Since power consumption is a scalar quantity, the reduction of power consumption of a component of a VE translates into the reduction of power consumption by the VE. In fairness, the translation is not direct (1:1). A generalization of Amdahl’s law comes to mind: improvement in a component, measurable by some metric, is attenuated by the ratio of that component’s use (measurable by that metric) to the system’s (the VE’s) use (measurable by the same metric). However, we can safely summarize the finer point we alluded to at the start of this paragraph as follows. Research that studies the measurement and modeling of power consumption by a component of a VE is in scope. We clarify this by example. For instance, we would:**Include** the Data Plane Development Kit (DPDK) [43], as it serves the critical function of networking (VEs that serve as virtual network functions (VNFs));**Include** a comparative study that measures power consumption by a VE using two different implementations of input/output virtualization technology, say: SR-IOV (single-root IO virtualization) and paravirtualization;**Exclude** a comparative study that measures power consumption by various network adapter (or network interface card (NIC)) architectures, **unless** it reveals the impact of these architectures on VEs’ power consumption.

Further detail emerges from the “real-time” requirement. This term is a reflection of the need for **application agnosticism**. Power meters that follow from such measurements and models may be used regardless of whether instantaneous or statistical readings of power are required. Studies (on measurements and models) meet this requirement by satisfying the following criteria.

1.Predictors:
a.Must be of fine temporal granularity;b.Must be updated with the regularity of the temporal granularity;c.Must enable prediction of power consumption at the same temporal granularity.2.Workload: Only resource-specific constraints are considered. That is, in the course of testing using, say, workloads that are processor-intensive (hence the workload is specific to the processing resource), no other constraints are allowed in works included in this study. In particular, models must not constrain the stochasticity of the workload.

Our last detail regards workload. We observe that the universal power model is a fallacy and the principal reason for this is that the interaction between workload and architecture cannot be pinned down indefinitely. This does not mean that modeling is a fruitless endeavor. It simply means that validity constraints must be placed on the model in terms of workload and architecture. Therefore, we do not exclude modeling and measurement because of its workload-scope or architectural scope. We do, however, observe that such models are pitfalls for those who apply them without knowledge of such limits.

We now proceed to present some recent, related surveys, highlighting their methodology and analytical approach. Against this background, we summarize the novelty of this survey.

### 1.6. Related Surveys

In recent years, much research has modeled the power consumption of servers in data centers and cloud environments in general. Several surveys in this regard cover different aspects of modeling and energy efficiency approaches for servers and virtual entities.

An analysis of power models the from micro- to macro-level is presented in [44]. This survey covers different aspects and levels of both hardware- and software-centric modeling techniques. Researchers studied models based on computing resources (CPU, memory, storage and I/O), system architecture (such as single or multiple cores), the availability of Graphics Processing Units (GPUs), system/network components, operating systems and virtualization environments. They categorize the existing models at different layers moving from architecture level modeling to the level of power models for whole data centers. However, the survey did not focus greatly on power modeling techniques that consider the effect of different virtual entities.

According to the categorization in [44], power models depend on different organizational contexts, including the power consumed by system components, running applications, and/or the execution strategy of processes. These metrics, however, can derive additive component-based, regression-based or machine-learning-based power models. Additive models usually present an aggregated view of server power consumption, which could be based on different resources (such as CPU, ram, I/O, disk), or the disaggregate static and dynamic power consumption of the server. Regression-based models are mostly based on the relation of power to the dynamic evolution of some measured system parameters. Power modeling using machine learning techniques is an advanced research area which can be further classified as supervised, unsupervised, reinforcement and evolutionary learning. Furthermore, at a much higher level, such as that of data center environments as a whole, power models for groups of servers, datacenter networks, conditioning and cooling systems were studied. The survey also analyzed the modeling methods at operating system (OS) and virtualization level, and for data-intensive, communication-intensive and general applications. At the end, the researchers compare the power models against their complexity, effectiveness, application and use-cases.

The survey carried out in [45] analyzes the power models based on their modeling approaches. It claims that power modeling methods can be divided into two main themes: analytical models and formula-learned models. In the former, parameters affecting the system power are known to the researcher; however, the weight for each parameter needs to be determined. On the contrary, in high-level formula-learned or machine-learning-based modeling, no prior knowledge of the system is required and the model is developed from scratch using the provided data set. The survey further describes commonly used machine learning techniques, and later focuses on the use and effectiveness of neural networks in power modeling. According to this survey, the approach to model power for any server may vary in several aspects. These different characteristics are summarized as (i) ***degree of autonomy***, i.e., to which level the power model is dependent on external hardware; (ii) ***level of granularity***, which describes the depth of logical (core, thread, etc.) or physical (device specific, system level, etc.) levels to which the model can precisely estimate the power; (iii) ***methodology***, describing the selection of method, which could be simulation, analytical modeling or data-training; (iv) ***simplicity***, which can be assessed by the number of variables, method selection and models’ computing overheads; (v) ***portability***, as most of the models developed are effective for some specific architecture, workload or environment, and their generalization is still a question; (vi) ***accuracy***, which defines the measurement accuracy or estimation precision of power for any model; and (vii) ***power meter*** (as most studies use external power meters as a ground truth for their modeling, their accuracy is also a major concern for developing a more efficient model).

The survey in [46] adopts a slightly different analytical approach and evaluates selected existing power modeling techniques in a unified environment. Comparative analysis has been performed for twenty-four different software power models and measurement methods, with nine different benchmarks under a single experimental environment. It evaluates the existing software power measurement techniques and models for different applications, benchmarks, systems configuration, server architecture and for their estimation errors. The authors claim that most of the software-based power models use system performance metrics provided by the operating system, or performance monitoring counters provided by hardware sub-systems of the server. Software power models considered in this study are categorized in three types as single variable CPU-based, multi-variable CPU-based, and single-variable throughput-based. The result of this unified experimental setup shows that power models based on a support vector machine (SVM) and interpolation techniques show the least error for different resource-intensive applications, whereas lasso regression with 30 variables was found to be the worst power model with the highest error. Furthermore, the modeling techniques are mainly divided into two categories; linear and non-linear, where each category is further classified based on its derivation from mathematical modeling or machine learning techniques. 

Another survey [47], with a more limited scope, focuses on the power modeling of servers in the cloud while considering the complexity stemming from the diversity of host hardware platforms, virtualization environment and workload. It presents the usability, applicability and limitations of the studied approaches, and also presents the analysis of the traditional existing and emerging modeling techniques. It reviews the existing power modeling methods at three stages: data acquisition methods, power consumption models and power modeling methods for servers, VMs and containers. According to this study, the collection of data for power models could be based on: (a) instruments such as external power meters, (b) dedicated acquisition systems which are generally products customized by developers for a specific hardware, (c) simulation-based, and (d) software monitoring tools. The latter constitute a widely used method that is based on system indicators and sensors. For the second stage—i.e., the power modeling—either hardware-centric, virtualization-centric or application-centric power modeling schemes can be used. Each type has its own drawback and usability; hence, the domain is still an open research area for further generalization and measurement precision. The selection of method, which is considered as the third step of the modeling in [47], is characterized as power-modeling-based on: (a) empirical parameterization, (b) function regression, (c) machine learning, and (d) evolutionary algorithms. The survey, after analyzing different power models, concluded that functional regression and machine learning methods yield high accuracy when provided with large enough data sets and clear power behavior of servers. This survey is more similar to ours; however, we distinguish further by providing statistical analysis of existing approaches, used tools, methods and measurement metrics. Moreover, our survey focuses on the modeling of power for individual virtual entities and on the effect of the virtualization environment on server and network functions’ power behavior.

In general, the surveys mentioned above present the literature on server models irrespective of the operational environment. One of the major trends, not only in data centers but also in networking, is the presence of virtual entities and the adoption of virtualization environments. In contrast, our survey concentrates on the power models of virtualized entities, and focuses mainly on components and parameters that could affect their power consumption. It further provides a statistical analysis of included research works with respect to their data acquisition methods, measurement tools and modeling methods, to give researchers a bird’s-eye view of the existing literature through which one can identify the approaches carried out in different studies.

### 1.7. Organization of This Survey

The survey is organized as follows:In [1], we describe our method: a problem–approach–development (PAD) triad, which, to the best of our knowledge, we are the first to use to identify research dynamics. We have delegated the description of the method to a separate paper, to resolve the difficulty of elaborating fully on the method without distracting attention from the results which we have obtained and which we document separately in this review.Section 2 presents the detailed results.Section 3 is our analysis of the results. There, we give a qualitative assessment through themes which emerged as we organized the data. We have classified these themes as “state of the art”, “fallacies” and “pitfalls”, to suggest guidance and warnings which we were able to glean from others’ experiences.Section 4 concludes by attempting to encapsulate the insights we have gained through this work.Appendix A illustrates the use of structural coding on a sample of the corpus on a popular research area: predictive models of renewable energy consumption in the radio access network.Abbreviations lists and expands the acronyms used in this paper.

We complement this paper with an online repository (https://github.com/humaira-salam/PowerMeasurementAndModelingRawData, accessed: 23 December 2022), that carries our raw data.

## 2. Survey Results: A Digest of Challenges, Approaches and Developments

### 2.1. A Taxonomy of the Problem Space

As our parsing of the literature proceeded (our method is described in [1]), we observed that the scope of this survey is relatively narrow and the problems in our set are not fully independent of one another. Rather, the problems diverge from one another only as ***aspects*** (we could also say that they are ***derivatives***) ***of the core challenge of modeling the power consumption of virtualized entities***. Each RU (research unit) (The RU, or unit of research, is “a publication (excluding surveys) in conference proceedings and journals” that “ha[s] three common manifest properties”, i.e., problem(s), approach(es) and development(s) [1]) is rooted in this core challenge, but the derivative problems (our *Problem categories* (P-categories) and their members) addressed differ from one RU to another. Figure 1 is an illustration of a simple organization of the challenges which have been tackled in the literature and shows their frequency of presence in RUs. The organization gives prominence to how challenges have been perceived:One group regards the concern with obtaining an understanding of the dependency of power consumption on some genre of artifacts. Categories P1,2 and P9–11 are in this group;The other group regards the concern with how to predict power consumption. Categories P3–8 and P12 are in this group.

We now proceed to describe the categories in more detail. Each description is preceded by a list of references to works that tackle a challenge in the problem category.

#### 2.1.1. Problem Category P1: Host System Hardware Architecture Perspective: Dependency of VE Power Consumption on Host System Architecture [48,49,50,51]

Challenges in problem category P1 address the impact of specific architectural attributes of the host system on the power consumption of VEs. They relate to changes in power consumption (the behavior) as major attributes of architecture and system-level designs are adjusted, inserted or removed. Insertions and removals are coarse configurative actions such as enabling or disabling; adjustments consist of progressive modifications such as adding increments of a resource. Examples of attributes which have been tackled include multiple processor cores, processor frequency scaling, Non-Uniform Memory Access (NUMA) and hardware threads (e.g., Intel Hyper-Threading). For network functions, the importance of knowledge about the power efficiency of NUMA and multiple-core architectures has the added relevance of these architectures’ relationship to ***determinism*** [37]. We dwell further on the underlying premise of ***hard partitioning*** in our consideration of the impact of the high-performance data plane on power efficiency (see Section 3.3.2).

Research that investigates the dependency of power consumption on architecture is ***exploratory, charting*** work. It attempts to provide a framework for detailed modeling through the discovery of broad relationships. Problems in this category arise with developments in architecture and system-level design. For example, while [49] is a comparatively old work that tackles architecture, [50] is newer and finds scope for research in system software’s exploitation of NUMA. One recent, highly significant scope is that of the use of ***domain-specific architectures (DSAs)***. Researchers are exploring specialized hardware in the quest for the improvement of the energy–performance–cost ratio, and will investigate energy efficiency in the process of their research. As domain-specific cores are mixed with general-purpose cores, many architectures will be investigated from each of the three pinnings: energy consumption, performance and cost. A particularly relevant set of DSAs regards real-time packet processing by computer systems hosting NFs at intermediate nodes (INs) at the network edge. Concern lies with expediting the common tasks, such as sending/receiving packets and processing headers. SR-IOV is a good example (see, for example [52], and its inclusion in [48]), but software-only solutions, such as poll-mode drivers, may also help to cut through the many middlemen characteristics of general-purpose operating systems [48,53]. Introduced to serve the perspective of performance, it is now necessary to understand their impact on power efficiency. Therefrom, it is necessary to understand how to ***control*** their power consumption. We suggest that “***profiling***”, the term chosen in [49,54], is a helpful descriptor of this kind of research. Just as a profile produces an external boundary within which to fill detail, so does this kind of research provide a framework through which modeling work is facilitated and within which modeling work provides details of power consumption relationships.

#### 2.1.2. P2: Impact of Alternative Virtualization Genres and Virtualization Platforms on VE Power Consumption [55,56]

Here, a behavior that consumes power is investigated across different ***implementations*** of a ***system concept***. We have observed investigation of two different system concepts: (a) virtualization genres and (b) virtualization platforms. There are three members of the virtualization genre group: containers, para-virtualization and hardware-assisted virtualization. In the virtualization platform group, examples include Xen, Hyper-V, Kernel Virtual Machine (KVM), Docker and Linux Containers (LXC). Research questions typical of category P2 seek to control the scope of experimentation through exercise of specific resources, e.g., per-host networking using emulated switches (software switches) [55,56], processor-bound and memory-intensive processing [54].

We consider genres and platforms as sub-categories of the same overarching problem category. Namely, this is system-level exploration that attempts to establish generalizations about an uncharted space. Like problems in category P1, new problems in this category arise with fresh alternative virtualization genres and platforms. However, here the scope of investigation is broader than with works classified under P1. Unlike P1, where specific architectural aspects (e.g., NUMA, hardware threads) are explored, the perspective taken here is a concern with the impact of the choice of implementation of a system.

#### 2.1.3. P3: Estimation of Power Consumption of (Virtualization-) Host System [57,58,59,60,61,62,63,64,65,66,67,68,69]

Measurement of a single server’s power consumption through the use of an external power meter is a trivial task. However, at the scale of cloud datacenters, it is a logistical burden. In addition, travel to the datacenter’s site may be burdensome. Furthermore, service availability would be reduced by the process of attaching a physical power meter to the hardware in the virtualization platform, e.g., between the server’s power inlet and the outlet in the racking cabinet’s power distribution unit (PDU) (naturally, availability would only be affected in cases that do not integrate (management and) measurement facilities within the PDU).

The alternative is the deployment of software power meters. In the scope of this survey, the cases we consider are meters that attempt to predict host power consumption on ***the basis of activity in the VE***. This challenge is tackled, for example, in [58,59,60,62,67,70]. These works then proceed to tackle the problem of attribution of system power to the guest VEs. Indeed, inclusion within the scope of both challenges (modeling power consumption of VEs and that of the host system power) seems to significantly enhance the usefulness of such research, with relatively less effort.

Host power consumption may be predicted in terms of VE resource utilization, or in terms of ***a simple characterization of the VEs’ workload***. The use of simple workload characterization as a predictor requires knowledge of workload parameters such as the number of processes, number of threads, web interactions per second and network interface utilization. Enokido’s and Takizawa’s work [67,68,71] is noteworthy in its consistency in modeling in these terms but other variants of this approach have been found: (a) web interactions per second [65] and (b) number of VMs running processor- and/or network-intensive workloads [66].

To contrast: works such as [57,72,73,74,75,76] are not included within this category, notwithstanding their development of models for the prediction of host system power consumption. In these works, host system models were developed as part of the scope of the challenge of modeling virtualized entities. Therefrom, the challenge of system power attribution (problem category P8) was tackled to proceed to guests’ power models.

#### 2.1.4. P4: Dependency of Power Model on Workload [56,62,67,68,69,71,75,76,77,78,79]

This category regards the perceived dependency of a VE’s power consumption model on the tasks it is processing. While it is intuitive to expect power consumption to depend on the workload, it seems far less intuitive to expect the model to depend on the workload. If this dependency is detected, the problem of model formation must undertake this aspect of investigation. Two different, major approaches towards achieving ***adaptability*** of the model to the workload have been observed:***Adaptation during run-time:*** the selected mode of instrumentation may not be suited to a generalizable, closed-form relationship between inputs and power consumption. In this case, model parameters must be re-trained online. This approach is therefore of the operating-time, or run-time, kind;***Off-line adaptation:*** a larger set of inputs may need to be identified to comprehensively characterize the variation of power consumption with the workload. This approach is therefore of the design-time, or off-line, kind.

We conclude this part with a note about two descriptors of the workload: homogeneous and specific. The term “homogeneous” is encountered in the literature to refer to the case where host system deployments within scope are subjected to a single workload. The term seems to originate in warehouse scale computing (WSC). Conclusions drawn from this kind of workload have drawn criticism as the results, while significant by virtue of the mass of WSC, are not generalizable. The other term—“specific”—identifies a single application; for example, a member of the Standard Performance Evaluation Corporation (SPEC) CPU2006 suite [80]. This term is used to indicate that models tested under such a workload are application-dependent and are valid only within a limited range of this dimension of variability (i.e., the “workload” dimension, see the treatment of the seven dimensions of variability).

#### 2.1.5. P5: Dependency of VE’s Power Consumption and Power Model on VE’s Resource Configuration (Heterogeneity) [62,73,78]

This category regards the perceived dependency of a VE’s power consumption and/or the dependency of its power consumption model on (a) the physical host’s resource configuration and (b) the individual VE’s resource allocation. Research here is concerned with two cases of very practical problems: the impact on power consumption of (a) the differences between hosting machines/containers and (b) the differences between virtual machines. We have observed that occurrences of research that undertake this challenge tackle it as an adjoint to another focus, not as the research’s primary objective.

***Physical host configuration:*** Host machines in a cloud datacenter may be expected to come in a limited variety of types, principally differing in resource capacities such as the number of processor packages per server, cores per processor package, amount of RAM per server, spread in storage device sizes, etc. Processor power consumption is notably variable, even within a single family of processors. Indeed, specialization in optimized power consumption within a family of processors is a part of the study carried out in [34] within the context of an edge cluster for use in NFV. As a VE migrates from one processor within a family to a processor of a different specialization, its power model will change.***Individual VE’s resource allocation:*** The power consumed by a VE varies with the allocation of resources to (i.e., in use by) a VE, which can be dynamically varied. The number of virtual cores assigned to a VE is a notable example, see, e.g., [57,61,72,73,74]. Moreover, VEs are commonly offered in sizes, e.g., small, medium and large, where allocation varies within all the major resource categories, demanding prediction of power consumption matched to the size of the purchased VE.

#### 2.1.6. P6: Impact of Temperature and/or Frequency on Models That Predict VEs’ Power Consumption [60,81,82]

This category regards the challenge of the inclusion of processor package temperature in models of power consumption. Works that tackle this challenge are concerned with detailed models of power consumption. Here, the interest lies in obtaining models that incorporate dependence on hyper-parametric attributes such as temperature.

#### 2.1.7. P7: Loading the VE’s Resources and Measuring Resource Use [56,57,58,59,61,62,65,68,69,71,73,75,76,77,78,79,81,82,83,84,85,86,87]

This category regards the use of computing resources and the measurement of such use by VEs. Interest stems from the role of resources ***as predictors in modeling***. The researcher is firstly concerned with ***loading*** (i.e., effecting the use of) resources. What means ***within the operating context of the VE*** can be used to load a resource? Should it be loaded in isolation (using synthetic loads) or should it be loaded using representative (realistic) workloads? Once these problems have been addressed, the concern with the ***measurement*** of resource use arises. The problem here consists of identifying the means that quantify resource use made by the loading.

#### 2.1.8. P8: Attribution of Host System Power Consumption to Individual VEs [57,58,59,61,62,73,75,76,77,79,81,82,84,86]

The attribution of host system power to individual VEs is a fundamental problem in proceeding from the directly measurable (host system power consumption) to the indirectly measurable (individual VEs’ power consumption). Direct measurement of host system power is possible (e.g., at the wall outlet, or through voltage rail in-line resistors), and such empirical evidence can be used as a ground truth and compared with power consumption inferred through modeling. How, then (and herein lies the problem), can this consumption be attributed to the host’s individual guests (the VEs)?

Within the host system, power consumption may be divided into ***idle (static), active (dynamic)*** and ***overhead***.

1.Idle (static) power consumption:
a.Power consumption while idle is not attributable to any VE at all, as this consumption arises out of the electronic behavior of semiconductor material, not of computation, communication or storage;b.Nonetheless, this power consumption must be accounted for and different approaches have been followed. For example, the physical machine’s idle power is attributed to individual VEs in fractions equal to the ratio of each VE’s virtual CPUs (vCPUs) count to the total complement of vCPUs active on the physical machine [57,72,73,74].2.Active (dynamic) power consumption:
a.The active component can be linked to a particular VE;b.This includes active power consumption in peripherals, e.g., network interface cards/adapters (NICs) and mass storage devices.3.Overhead, e.g.:
a.Operation of heat dissipating units (fans) to prevent thermal runaway;b.Losses in the power supply.

A “top” (host)—“down” (guest) approach to attribution has been observed.

1.Decide on what host system power consumption is within the scope of the study and how to divide it. The problem of attribution of the above three causes may be summarized as follows:
a.Is idle power attributed to the VEs or is it attributed to the host/a privileged guest?b.Is consumption by peripherals within the scope? How will this be attributed?c.Are overheads modeled or is correlation with other sources of power consumption going to account for them?2.Select a set of performance metrics that are correlated to a VE’s power consumption.3.Select a model that maps a VE’s performance metrics to its power consumption.4.This fourth consideration is tackled only by those researchers who investigate the ***adaptability*** of the attribution obtained through steps 1, 2 and 3. Does the obtained attribution adapt well to concurrent, co-hosted VEs? That is: if concurrent, co-hosted VEs were to be investigated, would the division, metrics and model still result in accurate prediction?

#### 2.1.9. P9: Implementation of Virtual I/O; P10: Implementation of Network Functions; P11: Implementation of Software Layer 2 (L2) Data Plane Switching [48,51,53]

These three categories are introduced together, since elements from the respective categories are commonly implemented ***as a set*** for the purpose of the realization of the virtualization of network functions. Here, researchers seek comparative statements and/or broad correlations (e.g., independent, positive, negative) between the workload (often in terms of packet rate and size) and power consumption, across implementations of the same type. As was observed for categories P1 and P2, researchers seek a profile of the power characteristics of implementations. It may be helpful to repeat that by “profile”, we understand that characteristics sought here are not of the detailed form of closed-form expressions. Examples of elements from the respective categories are:Virtual I/O (P10): virtio [88] and DPDK poll-mode drivers (PMDs) [89];Network functions (P11): Bro (now Zeek) [90] and Snort [91];Software layer 2 data plane switching (P12): Open Virtual Switch (OvS) [92] and VALE [93].

Problems in each of these three categories merit separate classification as they have been tackled separately in the literature. For example, in [53], a number of components are investigated: three different implementations of software virtual switch (P12), two different I/O virtualization devices and two different implementations of the same network function (intrusion detection system (IDS)). In [51], power consumption by packet transmission under DPDK is investigated under the condition of enforcement of (a) the network adapter’s affinity to NUMA nodes and (b) DPDK process pinning to processor cores.

#### 2.1.10. P12: Investigation of Processor Green Capabilities [50,60,67,81,82,83,85,87,94]

Works in this category investigate the low-power idle (LPI) and adaptive rate (AR) operation of a processor as a means of reducing power consumption. The challenge is broad enough to permit a sub-categorization into (a) those works that investigate the influence of frequency as a power-model parameter [81,82] and (b) other works that address improved, real-time governance of LPI and/or AR [83,87,94] to minimize the power consumed to process a load.

#### 2.1.11. P13: Improvement of Power Efficiency of High-Performance IO Virtualization Frameworks [87]

A separate classification was set up for [87] as this work represents an evolution of those classified under P9. This work extends beyond profiling and suggests use of low-power instructions as the means to balance performance and power efficiency.

### 2.2. A Taxonomy of Approaches

Figure 2 illustrates the taxonomy we use to structure the approaches detected in research work. Line thickness and percentage values represent the ***utility*** of the specific approach. Utility is best understood within the context of all the observed triads in a literature corpus. For any specific approach, this may be used to solve a variety of problems and its application may result in a variety of developments. One may therefore think of the approach as a ***nexus***, or a point of confluence through which many researchers pass as they attempt to solve problems. Thereafter, researchers diverge radially outward from this point of confluence towards some achievement (some development). A suggestive image is to think of the approach as the center of a star, but spokes converge onto it from problems and diverge away from it onto developments. When the count of these triads (each composed of two radial lines, or dyads) is divided by the sum of all such counts for all approaches, we obtain a metric: a normalized quantity obtained within the context of all approaches. See our method [1] for the formal statement of ***utility***.

We now proceed to describe the categories within the context of the taxonomy. Each approach-category’s description is preceded by a list of references to works, each one of which uses a component within that category’s set.

#### 2.2.1. Analytical Foundations

This group of categories regards the theory and hypotheses that comprise the essential abstractions at the basis of scientific research.

**A1: Power attribution principle** [48,53,57,58,59,61,62,69,73,75,76,77,79,81,82,84,86]: When ***host system*** power is measured, whether at the wall outlet or at one or more of the power supply’s output lines, there is the problem of attributing the measurement to the logical divisions (VEs) of the host computer system. The attribution of system power to modeled entities starts with a decision on which power consumption is within scope (see Section 2.1.8). Next, power in scope is attributed to (burdened on) one or more entity. For example: will idle power consumption be attributed to the host system or will it be attributed to the VEs?

**A2: Modeling bias** [57,58,59,60,61,62,63,64,72,75,76,77,79,81,83,84,85,87,94]: Researchers approach the problem of developing a model under some bias which conditions their final outcome. This bias is manifest in researchers’ selection of a particular type of regression to apply to their data. We note that this same observation is carried in [95]. Here, our purpose is solely to draw attention to what we have observed as researchers’ **modus operandi** without analyzing their choice of approach.

**A3: Green operating principles** [83,87,94]: Works in this category weave radical approaches to power efficiency into their developments. For example, instead of conventional scheduling, ***run-to-completion*** [96] is exploited to obtain dedicated (or, at least, very sparsely shared) resources for the processing of packets. This approach is further nuanced by the real-time control of adaptive rates and sleep depth. In one particular case [87], the novel concept of a low-power instruction instead of transitions to/from low-power idle (sleep) states is used.

**A4: Physical analysis** [60]: This category regards approaches rooted in the physical properties of (semiconductor) material in the consumption of energy. Only one work [60] was found fitting this category. However, another two that used this approach to study the power consumption of physical entities (and therefore was outside the scope of this study, which is concerned with VEs) were found and they are described next to illustrate the approach better. In [70], a study implicitly applies Dennard’s law in the process of obtaining weights that scale a processor sub-unit’s contribution to power consumption. In [97], the physical cause of power consumption in metal-oxide-semiconductor (MOS) material is examined and used as the basis for modeling equations.

**A5: Identification and use of metrics of energy efficiency** [49,50,65,85]: The relationship between system architecture and power consumption can be investigated through the identification of the relevant metrics of energy efficiency. For example, an easily recognizable metric, albeit rather broad in possible interpretations, is the J/b (joule/bit). The use of such metrics encourages joint consideration of function and power consumption.

#### 2.2.2. Experiment Design

The practical, hands-on aspects of the empirical process are the product of a (probably cyclical) design phase, concerned with a number of stewarding activities pertinent to test subjects and ancillary objects in the testing scenario, instrumentation, inputs and outputs. The activities include selection, configuration, interconnection, initiation, observation and termination. We have identified several examples of such activities within our research scope and grouped them under ***resource provision*** (categories A6, A7), ***workload selection*** (A8, A9) and ***data collection*** (A10–A13). We describe these categories next. Admittedly, the activities referred to (i.e., selection, configuration, etc.) have broad meanings; therefore, in the course of describing the categories, references to the activities are emphasized by bold, italicized text.

**A6: Managed resource provision** [48,49,50,51,53,55,56,60,66,67,68,73,81,82,83] **(*selection*, *configuration*):** This concerns the provision of resource capacity either to a specific VE (the guest system) or to the physical entity (the host system) hosting the VEs. Within the empirical process, the techniques in this category provide the means to observe the effect on the power consumption of managed changes in resource provision. Examples include the (manual) pre-***configuration*** of:The frequency of operation of processor cores [48,50,81];Core affinity [51,55] and hardware-thread affinity [68];The network interface data rate capacity capping [66].

These techniques are executed as part of the process of ***selection*** of the operating parameters, i.e., setting up experimentation, ***before*** operations start. This qualification is necessary to distinguish from such approaches as may change the operating, run-time context.

**A7: Controlled resource provision** [78,83,85,87] ***(configuration)*:** Provision of resources may change during the running of an experiment (rather than before it starts). Approaches in this category include the automated adjustment of:Processor frequency (also known as performance state, or P-state) [78,83,85,87];The depth of processor sleep (also known as low-power idle state, or C-state) [83];The number of hardware threads [78];The time spent running a low-power instruction [87].

These techniques are approaches to solving the problem of full-throttle operation. Without a guided operation of adjustments such as those listed above, operation of the processor may quite reasonably be likened to a multi-assembly-line manufacturing plant that operates line machinery whether there are goods to produce or not.

**A8: Resource-specific workload, A9: Representative workloads *(selection, configuration, interconnection)*:** These two categories regard the ***workload selection*** stage, within experiment design in the scientific method. The workload comprises the ***inputs*** referred to earlier; inputs must be ***interconnected*** to the system under test, and this is often not a trivial task. In our thematic analysis in Section 3, we identify ***workload type*** as one of the seven dimensions of the variability of power models. The influence of workload type on the model obtained is evident in the attention paid by researchers to their selection of workload type. We can distinguish two broad categories of type.

**Resource-specific workloads (A8)** [48,49,50,51,53,55,56,59,60,61,63,64,66,67,69,71,75,76,77,84,86] are applied to investigate the impact of the utilization of specific resources on power consumption. Such ***synthetic*** workloads are applied ***(interconnected)*** to a machine (whether virtual or physical) to reduce (as much as possible) the scope of power-consuming resources to a targeted set. Resource-specific workloads are most commonly used in exploratory work, to gain an understanding of the behavior of a resource’s power consumption. We refer to this approach as resource-specific workloads, synthetic workloads or resource isolation.

**Representative workloads (A9)** [48,56,57,58,60,62,65,68,69,73,75,78,79,81,82,83,84,85,87] may be used as complementary with, or alternative to, resource-specific (synthetic) workloads. A notable complementary use is made in the testing (post-training) phase of model development, when representative workloads are used to validate a model (obtained using synthetic workloads). They may also be used in an entirely alternative approach to synthetic loading, to support development of application-agnostic models. Representative workloads lead to training data that incorporates variation in utilization of more than one resource at a time; hence providing at least limited application agnosticism.

The next four categories (A10–A13) regard the ***data collection*** stage, within experiment design in the scientific method. Categories A8 and A9 regard the ***selection*** of workload type. The approaches described here regard measuring ***how much*** of a resource is being used, or workload is being applied, and how long to apply the workload to obtain statistically valid results ***(initiation and termination)***. Categories A10, A11 and A13 regard instrumentation ***(observation)*** of those variables considered (by the researchers concerned) to be reliable predictors of power consumption.

**Resource instrumentation in microarchitecture and system software (A10)** [48,49,56,58,59,61,62,63,64,69,73,75,76,77,78,79,81,82,83,84,85,86,87] includes approaches that measure resource use. These measurements are then used to predict power consumption. We make a somewhat weak distinction, for reasons we shall refer to, between instrumentation of microarchitecture and instrumentation by system software. The former regards parts of the processor interface that address the processor’s infrastructure for monitoring, event counters (e.g., instructions retired, last-level-cache (LLC) misses and translation lookaside buffer misses) and, more recently, power counters (e.g., Intel’s Running Average Power Limit (RAPL)) [98]. On the other hand, system software’s instrumentation is carried out through intermediary system software and includes, most notably, processor utilization. We have seen references to these two categories as “hw counters” and “sw counters”, respectively [99]. The distinction is weak since system software is increasingly exposing microarchitecture instrumentation (consider, for example, Linux’s ***perf*** tool). This reduces the need to directly access hardware registers and blurs the separation between what is abstracted and what is concrete, raw, hardware data. Nonetheless, through our understanding of the data used, we have been able to separate the approaches into two sub-categories.

**Category A11 regards the use of a simple characterization of workload as a predictor of power consumption** [48,49,53,64,65,67,68,71,78,83,85,87]. This is notably different from approaches in category A10, which are concerned with resource use as a predictor of power consumption. Other examples (apart from those given earlier) of workload metrics as inputs are the “number of processes” [71] (from the same process image), “transmission rate” [68] and millions of instructions per second (MIPS) [64].

Characterizations may need to be sharper. For example, since packet network traffic arrival is known to often have the properties of a Batch Markov Arrival (stochastic) Process (BMAP), this is an operating constraint (***selection, configuration*** and ***interconnection*** are all ingredient activities here) adopted in the approach of several works studying power efficiency of network functions [83,87,94,100].

**Category A12 refers to the measurement of (host) power consumption** [48,49,50,53,55,56,57,58,59,61,62,63,64,67,68,71,73,76,79,84] ***(observation)*** which is most usually measured at the wall outlet or at the power supply inlet. More granular approaches are desirable, and indeed we do find cases [62] that attempt measurement at the power supply output. The principal drawback of such granular techniques is not (principally, at least) construction of intermediary hardware (e.g., riser boards or line resistors) but the difficulty in attributing power drawn through any single dc voltage output (or group thereof) to particular consumers. With the advent of RAPL and certain guarantees on its accuracy, the need for direct power measurement has been, at least partially, avoided.

**Category A13 regards the use of hardware sensors to obtain inputs and/or parameters for the power model** [60,81,82,87]. These include:Voltage sensors (processor supply voltage);Temperature sensors (processor package and memory temperatures);Fan speed sensors (processor and chassis fans);Wall-clock time measurement.

Some of these variables are used in models that predict power consumption while accounting for the effect of the drift of temperature and automated supply voltage adjustment (in dynamic voltage and frequency scaling—DVFS).

#### 2.2.3. Model Validation

Model validation is a multi-faceted endeavor and this is reflected in the approaches we have detected. The approaches range across the candidates that would typically be considered: simulation (A14) [87], use of test data (A15) [57,62,63,64,77,85] and corroboration through experimentation (A16) [83,87]. We skip elaborating on these categories as they are either self-evident (A15) or because they are too rarely used to permit general commentary. However, to these three categories we add a fourth (A17), namely, the model adaptation technique, which we describe below and explain why it fits within this branch of the taxonomy.

**Model adaptation technique (A17)** [57,58,59,61,62,67,68,69,71,73,75,76,78,79,81,82,84,86]: This refers to the approach(es) taken (if any) to develop an adaptable model or modeling system. Here, ***adaptability*** refers to the fitness for use which the model exhibits under changes in one or more of the seven dimensions of variability that will be defined in detail in Section 3.1.1 below. Model adaptability is essential for practical virtualization, where changes in, for example, the number of co-hosted, concurrent VEs, or in workload type, are commonplace. Here, we list the major approaches taken towards producing adaptable models. Since these approaches emerge in the context of validating a model’s accuracy under some limited range of the seven-dimensional space of operating conditions, we classify this category of approaches as an aspect of model validation.

***Adaptation to change in the number of co-hosted, concurrent VEs*** is widely achieved through the ***time-division multiplexing*** of event counters [59,62,69,79,86], RAPL counters and CPU utilization [61,73]. This approach enables the use of such metrics as predictors of dynamic (active) power consumption, by apportioning counts to VEs in accordance with the time during which the VEs were active.***Adaptation to uncorrelated causes of power consumption*** can be achieved through ***additional predictors*** [82] to follow causes of power consumption that do not correlate well with counters within the current set. This case reflects itself as poor accuracy in predicted power consumption. Although counter-based models are reported to fit a variety of processor- and memory-intensive workloads well, it may be necessary to account for unanticipated activity through the approach of adding previously unused counters.***It may not be possible to fit a single model with parameters known* a priori**, to the whole range of inputs within the scope of study, notwithstanding the diversity of predictors employed in this pursuit. The following adaptive techniques have been found in the literature.
a.***Dataset partitioning***, where the dataset is of the form { {predictors}, response } and is used in [77] to match the best model out of a set of models to the current, actual operation. An individual model in the set is associated with a single node in a decision tree and the node is selected according to features pertinent to the current, actual operation. A simpler, but conceptually similar, approach is taken in [58]. A number of models are devised and model-selection features are limited to the number of active VMs and a coarse grading of CPU utilization.b.***Modeling on demand*** is the term we use to succinctly refer to the fourth adaptation class of techniques:
i.One early example of this approach is found in [75], where the dependence of the model on the workload has been addressed through ***online training***, whenever prediction accuracy of the extant models falls out of a range of tolerance. The rationale adopted is that if model adaptation to such an “unseen” case is limited to parametric tuning, then a modeling system might be able to construct a model while VEs are in operation;ii.A broader perspective is found in [78]. An automated system for profiling containerized applications is described and demonstrated. Containerized applications are profiled from three perspectives: computing resources consumed, energy consumed and performance. In this case, the rationale is that energy consumption can be optimized by the determination of a frequency-and-hardware-threads host configuration that meets performance requirements. Thus, starting from central functional requirements (performance requirements), operating conditions are determined that minimize energy consumption. This approach is capable of meeting the challenges posed by heterogeneous host hardware and application (workload) diversity, at the cost of analytical modeling. Indeed, characteristic curves can be derived but causes underlying observed behaviors remain unaddressed.

### 2.3. A taxonomy of Developments

Developments fall cleanly into one of two groups: (a) models of power consumption and (b) observations on dependencies of power consumption. The first group (D1–D10) includes developments that ***predict power consumption*** over a sub-space of the seven-dimensional space of operating conditions. The second group (D11–D18) includes developments that are oriented towards the ***correlation of power consumption*** with aspects of system integration. As the taxonomy is rather broad, we present it in three parts:Figure 3: top-level fork into models and dependencies;Figure 4: the taxonomy of models;Figure 5: the taxonomy of dependencies

**D1–D4:** We first present four categories of developments that concern ***models of host-system power consumption*** characterized by the condition where ***workload is processed by VEs***:D1: linear regression models;D2: non-linear regression models;D3: machine-learnt models;D4: models of local mass storage.

This contrasts with the scope of developments referred to under categories D5, D6 and D7, where ***models of VE power consumption*** are presented. 

Models in these categories are interesting from the perspective of analyses of ***sets of hosted VEs*** that seek to identify operating conditions of optimal host power efficiency. As predictors, such analyses use instrumentation that measures resources used by the VEs. Categories D1, D2 and D3 all predict power consumption in terms of resource use but differ in the type of model produced.

D1 regards models of power consumption through linear combinations of scalar predictors [57,58,59,69,75]. The scalar predictors are resource usage metrics.D2 regards polynomial or simple mathematical powers of resource use (the scalar predictors) [57,60,61,76,81].D3 regards models that employ machine learning (e.g., Gaussian Mixture, Support Vector Machine, Neural Networks) [62,77,79].

Category D4 regards the models of power consumed by mass storage local to the host system [86]. These models attempt to predict power consumption in terms of activity metrics such as the total amount of time spent in a known state (in terms of power consumption, e.g., active/idle), the rate of data exchange (MB/s or input/output operations per second) and the mode of operation (sequential/random and read/write). In the context of the approximations observed in the development of these models, their accuracy cannot be fair.

#### 2.3.1. Models of Power Consumption

**Adaptable VE models (D5–D7):** Developments within these categories consist of ***adaptable*** models of the virtualized entity’s power consumption and have two important characteristics in common:They are adaptable to a variable number of concurrent, co-hosted (active on the same host system) VEs (we refer to the latter scope of variability as the seventh dimension of variability);The predictors are the measured amount of computing resources used by the VEs.

Models can be distinguished by the predictors they use, workloads employed and modeling approach:1.***Predictors*** (of VE power consumption) are obtained from system software’s instrumentation, e.g., CPU utilization (see approach categories A10–A13), and from microarchitecture instrumentation, e.g., LLC misses (again, see approach categories A10–A13);2.***Workloads*** used to obtain the model (this restricts the range of workloads within which the model is valid) may be:
a.Specific workloads: the most restrictive, as they relate to a particular test load;b.Synthetic workloads: less restrictive than specific but limited to the exercise of one resource, typically the CPU;c.Combinational workloads: still less restrictive, involving the exercise of a number of resources of the host system (e.g., SPEC CPU benchmarks may be both processor and memory intensive);d.Representative workloads (e.g., TPC-W [101]) produce models that are readily associated with use cases.3.The modeling approach may be:
a.Linear regression (category D5) [58,59,69,73,75,81,82,84,86];b.Power (integer- and non-integer powers), exponential and logarithmic regressions (category D6) [61,66,76]; c.Machine-learnt combinations of resource use (category D7) [62,79].

**Models of power consumption that use workload profile as predictors (D8, D9**): Categories D8 and D9 group developments from (two) sets of RUs that predict power consumption of hosts and/or VEs through (the measurement of) some characteristic of the submitted workload. This contrasts with RUs in categories D1–D7, where prediction is obtained through (the measurement of) some computing resource (processor and/or memory and/or I/O). Most developments in this category are obtained through the abstraction of hardware by one or more model parameters that express power consumption under case-specific conditions of operation. Some of these abstractions are identified in the descriptions of these two categories.

**Developments in D8** [63,64,65,66,67,68,71,78,83,85,87,94] use:Processing load (number of processes, millions of instructions per second (MIPS), etc.) pertaining to a specific application, as predictors of ***host system*** power [63,64,65,66,67,71];Packets per second, through an intrusion detection system implemented in a VNF [85];Transcoded frames per second, through a transcoder implemented in a containerized network function (CNF), and inferred images per second, also in a CNF [78];Average network transmission rate, as a predictor of host system power [68];Statistics of a Batch Markov Arrival Process (BMAP) (packet traffic) as a means of the prediction of power consumption by a VNF [83,94].

Hardware is abstracted through the measurement of power consumption at some operating point (a specific operation is being carried out), or a change in power consumption over some operating range. Examples follow:In [65], where energy efficiency of an interactive web service is studied, the operating point is an entire VM running the TPC-W benchmark [101].In [67,71], the operating points are the host’s power consumption when (a) idle, (b) one core is active (processing load) and (c) maximum, with all cores active. Furthermore, use is made of the step increment in consumption corresponding to the activation of each additional core. Cores are activated when they are utilized by VEs.In [68], the operating point is the power consumption when co-hosted VEs are transferring a file to a client computer. An ***affine*** relationship between the host’s power consumption and its transmission rate (transmissions originate on hosted VEs) is found.In [64], an operating range is used: the increase in power consumption that corresponds to an increase in MIPS on the VEs.In [83], the operating point is the power consumption when a VE running on a single processor core is switching packets at the maximum rate for a given performance state.In [85], fifty-four (54) different features of network traffic are input to an artificial neural network that selects the operating frequency that optimizes power consumption.

With one exception, none of the works in the above list uncovers the hood to peer at the processor’s internals (to obtain predictors of power consumption). The exception is [83]; yet even in this case, the performance monitoring counters are not used as direct predictors of power consumption, but to obtain (a) the timing information necessary for a queueing model and (b) the operating state (ACPI (Advanced Configuration and Power Interface [102]) P- and C-states).

**Developments in D9** [87] are set within the models branch of the taxonomy. These developments may be considered as useful observations on ***the operation of processors’ green capabilities***. Examples of these observations (all from [87]) include:Low-power instructions might be a better candidate than low-power idle to save power under higher link utilization;Operation in full ACPI P-state, operation with low-power instructions on idle detection and operation with low-power idle on idle detection are (a) in ascending order of latency to return to active processing and (b) in descending order of power consumption;A processor utilization threshold below which low packet latency is guaranteed under BMAP traffic arrival is 80%.

By “green capabilities”, we refer to a broader range of microarchitectural aspects than the by-now-conventional adaptive rate and low-power idle operation. While these latter two remain at the center of attention, there is also the means of low-power instructions [87] that has been successfully employed to improve power efficiency. Notwithstanding the origin of these observations in modeling work, it may be argued that they might also be classified within the dependencies branch of the taxonomy. We have chosen the models branch, but as further studies add to the body of knowledge on how to operate processor green capabilities, this category’s position in the taxonomy might need to be changed.

**Relative accuracy of modeling approaches (D10)** [60,62,63,77]: Developments presented under category D10 are comparisons of the relative accuracy of alternative modeling methods with respect to conventional polynomial (including linear), power, exponential and logarithmic regression. These developments have been found within works that show models classified under category D9. The purpose is to qualify and quantify improvements of machine-learnt models with respect to conventional regression models.

#### 2.3.2. Dependencies of Power Consumption

This parent node of the taxonomy is divided into two child nodes that are not strictly mutually exclusive. For example, the software data plane is considered in works under D15. Clearly, the software data plane is a logical artifact and might be included within a child node of “physical and logical artifacts”, or directly thereunder as a leaf node (i.e., as a category). The choice of separation of D15–D18 and the inclusion under the parent node “networking workloads” was taken for two principal reasons. Firstly, the recurrence of investigation of power consumption’s dependency on networking workload merits attention through separation. Secondly, as this survey caters for an audience with an interest in softwarized networking, an emphasis on the power dependency on networking workload seems justified.

**Knowledge about the dependency of power consumption on specific hardware (D11); the dependency of power consumption on architecture (D12); the dependency of power consumption on resource provisioning (D13); and the dependency of power consumption on virtualization genre and technology (D14):** Categories D11–D14 are grouped into a set of works that obtain the sense of the correlation (positive/negative/none) between power consumption and some genre of artifacts:Specific hardware types (D11);Computer architecture (D12);Resources provided (D13);Virtualization genre and technology (D14).

Category D12 groups works that relate to observations on the impact of architectural features on power consumption [49,50,67,71,76]. While these observations are useful, they are generally too coarse to be directly applicable to real-time power control. Their use emerges from guidance which they provide in the development of power models. For example, it was observed that when the number of threads that fully occupy a core’s time (active 100%) exceeds the number of logical cores in the system, the energy efficiency (measured, in this case, in hash/J, a computational metric of energy efficiency) decreases [50]. Evidently, this is good guidance; equally evidently, it is not a directly applicable development.

Categories D11, D13 and D14 relate closely to (various aspects of) implementation, and as such are of particular interest to the ***system integrator***. Data of high quality from these developments inform and guide the tasks of gathering components into systems that meet the non-functional requirements obtained from concern with energy and power efficiency.

Category D11 gathers observations about the dependency of power consumption on specific processor hardware [49,69,76]. We have observed that these developments are gathered as a by-product of the process of research; they are rather incidental. Like any implementation, their usefulness is limited to the lifetime of the concerned device(s).

Category D13 [73] includes developments that regard the specific resource configuration of:The instantiating host, i.e., the relationship between a VE’s power consumption and the specific resources of its host hardware specifics such as the number of cores and amount of memory carried by the host instance;The guest VE, i.e., the variation of a VE’s power consumption with its resource assignment, on the same host.

Category D14 gathers observations about the dependency of power consumption on instances of virtualization genre and technology [55,56]. Developments in this category are less incidental than those in category D2 and are obtained with the focused intention of tackling challenges relating the power consumption of implementations. These developments relate to a less diversified group of implementations. For example, there are fewer virtualization platforms than processors to choose from. Direct use of these developments is mostly limited to the specific implementations concerned; however, some generalizable conclusions exist. For instance, it was observed that both hardware-assisted virtualization and paravirtualization are less efficient (in the specific empirical setup) than non-virtualized operation in the use of processor caches [56]. This empirical evidence favors the hypothesis that cache hit ratios suffer due to the greater thread rotation in virtualized and containerized environments.

**Knowledge about dependency on power consumption while processing a networking workload (D15–D18)**: Development categories D15, D16, D17 and D18 are grouped here as their central characterization is knowledge about the behavior of power consumption by VEs while processing a networking workload. Categories D15, D16 and D17 reflect the challenges described in P11, P9 and P10, respectively.

Category D15 regards contributions to knowledge about the power consumption of software data planes [48,53]; D16 regards virtualization of network I/O [51,53], and D17 is about network functions [48,53,85]. Thus, in [51], the power efficiency of DPDK PMDs is demonstrated with respect to Netmap drivers, for packet transmission. This development is balanced by [48], where the power efficiency of transmission through a DPDK-enhanced Open vSwitch is shown to be worse (@500-byte Maximum Transmission Unit (MTU)) than that of the unenhanced Open vSwitch. In each of these categories, efforts are made to allocate burden through the isolation of power consumption and the attribution to the sub-system (data plane/virtualized IO/network function) under study.

Category D18’s developments differ from those of D15–D17 because they cut across these categories’ sub-system boundaries [56,65]. For example, in [65], the energy efficiency of web transactions executed on co-located VMs is found to be highest in the operating condition of processor-core over-subscription (more VMs than cores). In [56], it is shown that power consumed by packet delivery to a VM through a software packet switch is much higher than that required for delivery in a non-virtualized environment. In both these papers, the object of interest incorporates the virtualization of network I/O ***and*** the data plane. In [65], the object of interest encompasses the network function: a web service and accompanying application and database components.

All four categories are of keen interest to the system integrator. The emphasis is on the components in the integrator’s set of building blocks, specifically on the behavior of power consumption of various implementations, and types thereof (in the scope of the categories).

### 2.4. P–A Dyads (Problem/Challenge-Approaches) Graphics

Over the following pages, we present a comprehensive set of graphics (Figure 6, Figure 7, Figure 8, Figure 9, Figure 10, Figure 11, Figure 12, Figure 13, Figure 14, Figure 15, Figure 16 and Figure 17) that illustrate the approaches detected to tackle specific challenges (reference numbers for pertinent RUs appear in the shape tags). For example, consider Figure 7. The graphic shows the component approaches applied to discover “the impact of specific architectural attributes of the host system on power consumption of VEs” (see problem category P1). 

The presence of approach categories A1, A5, A6, A8 and A10–A12 does not mean that ***every*** RU tackling this challenge uses a component from ***all*** of the approach categories. It does, however, mean that every RU tackling P1 uses a subset of the approaches shown. We proceed by giving examples (relevant to P1), with references.

***Managed resource provision (A6):*** Prior to running experiments, researchers set the conditions for the experiment through this approach [48,49,50,51].***Resource-specific workload (A8):*** This may be used to stress the component implementing the architectural attribute under test [48,49,50,51].***Simple workload characterization and instrumentation (A11):*** The workload must be characterized by some parameter that serves to measure its demand for power [48,49].***Resource instrumentation (A10):*** This is an alternative to the use of workload profiling (A11) as a predictor of power consumption. Rather than use, say, the number of threads, or transmit bandwidth (for networking workloads) as predictors, this approach uses resource instrumentation [48,49].***Identification of metric of energy efficiency (A5):*** In certain cases [49,50], energy or power efficiency is investigated, rather than energy or power consumption. In these cases, the researchers identify and use a relevant metric of efficiency, rather than metrics of consumption (watts or joules).

On any of the dyad graphics, the approaches shown (inside the approach categories) include only those which are used by at least one RU that tackles the challenge category that is the root of the dyads. We delegate a repository with a full tabulation of all approaches in any approach category. Similarly, all individual developments within a category are delegated to the same repository (https://github.com/humaira-salam/PowerMeasurementAndModelingRawData, accessed: 30 October 2022).

### 2.5. Causality DAG

The Causality Directed Acyclic Graph (DAG) in Figure 18 shows a bird’s eye-view of the proceedings of research in scope.

### 2.6. Triads (Problem/Challenge-Approaches-Developments) Graphic

Figure 19 shows the triads graphic. To improve readability, we limit illustration to the triads that are in the top twenty percentile of a list ordered according to thickness. These triads comprise 49.2% of the total number of compiled triads.

### 2.7. Statistics

Bar charts that illustrate the category metrics described in [1, N. see section on “Statistics”] are presented below:1.Challenges (Figure 20)
a.Frequency of occurrence in the set of all RUs;b.Frequency of occurrence in the set of all challenges in all RUs;b.Frequency of occurrence, weighted by approach diversity, in the set of all challenges in all RUs.2.Approaches: (Figure 21)
a.Frequency of occurrence in the set of all approaches in all RUs;b.Frequency of occurrence in the set of all triads in all RUs. 3.Developments: frequency of occurrence in the set of all developments in all RUs: (Figure 22)

To improve legibility, the x-axis labels show category numbers only. The codes (“terse, dense representations of a verbose articulation of a concept”, see [1, N. see sub-section “what are codes?”]) linked to the numbers are shown in Table 2, Table 3 and Table 4. In Table 2, we also show questions that help to clarify articulation of the challenge posed.

## 3. Thematic Analysis

The themes presented in this section are the product of a thematic analysis undertaken according to the method described in [1]. We first present an overview through a graphic (Figure 23) that groups the themes and then proceed to an exposition of the themes within the sub-sections.

### 3.1. State of the Art

#### 3.1.1. Seven Dimensions of Variability

Through our collation and resolution of the derivatives of the core challenge in this research space, we observe seven dimensions of variability in modeling power consumption by a virtual entity. Comparison with the power consumption of physical machines throws the core challenge into sharper relief. With physical machines:Power consumption can be measured directly;There is no virtualizing agent to consider;The activity of other physical machines (that do not send or receive workload) is irrelevant.

The seven dimensions of variability are shown in Table 5. It is not surprising that the problems we have seen researchers tackle are closely aligned with these dimensions. The research space is precisely about the need to obtain an understanding of the impact that these variables have on power consumption. The scope of models we found in the RUs covers only a subspace of the seven-dimensional space but the extent is not usually stated. We touch upon this issue briefly in our treatment of the pitfall of research without context and the fallacy of the universal power model.

#### 3.1.2. Adaptable Models

We observe ***three conditions*** which must be met for an automated modeling system to obtain a model of VE power consumption.

The first fundamental condition is common to all successful research into modeling of VE power consumption. ***Resources utilized by the VE*** (whether measured through architectural or microarchitectural instrumentation) ***must be strongly correlated to power consumption by the VE as well as power consumption in host system overheads***. If the overheads are uncorrelated to the VE’s activity, or weakly so, then any significant power overhead must be modelled through a separate consideration of its driving causes, e.g., temperature [81].

Furthermore, in order that modeling may apply across a diversity of operating contexts, two other conditions must hold.

Any change in the parameters of correlation must be dynamically learnt and the model adjusted.Any change in operating context that invalidates the parameters of correlation must be of finite duration. An indefinite transient precludes the formation of a model.

To this observation on the three conditions, we add another observation. Two of the seven dimensions of variability are commonly investigated in terms of the validation of the accuracy of modeling systems: ***workload type and co-hosted, concurrent VEs***. We suggest that a modeling system may be labelled adaptable if, minimally, ***it meets the above three conditions under these two dimensions, i.e., (a) operations of variable workload type per VE, and (b) a variable number of concurrent, co-hosted VEs.*** Before proceeding to refer to validations observed in the RUs, it is useful to draw attention here to the need for a ***modeling system***, rather than simply a model, in the estimation of power consumption. Dynamic adjustments are affected through the intervention of such a system, which adjusts model parameters to the operating context. We now proceed to describe how the variables in the two dimensions were varied in some studies.

**Workload type:** Different types of workload correspond to different use of resources; hence, the behavior of power consumption also differs. Several researchers [48,75,76] studied the effect of changing workload type either by categorizing the workload itself or by categorizing the resources consumed by a specific kind of workload. Both [75] and [76] characterized the use of CPU utilization for workloads, which is known to have a workload-type dependent relationship with power consumption [62]. Thus, in both these studies, the need arises to re-train on the change of workload type (second condition). Further, both carry an observation about the duration of re-training for refined models (third condition). 

On the other hand, each one of [62,69,79,81,82] is capable of adapting to different workload types without re-training. All predictors here are event counters. However, events are not necessarily linear predictors of power consumption across all workload types. In [84], it was found that the model parameters of a linear regression of event counters onto power consumption are workload specific. Given the contrast with [73,74], it seems that the root cause is the selection of events for prediction.

**Concurrent operation:** In [61], a non-linear model of the dynamic power of a multi-(virtual)-core VM is obtained. The dynamic power $p_{vm}$ is expressed in terms of the average utilization $u_{v}$ which *n* virtual cores impose on a total of *N* physical cores:$$p_{vm}\left(u_{v},n\right)=\alpha {\left(\frac{n}{N}\right)}^{\beta }u_{v}^{\beta }$$

Parameters *α* and *β* are determined through linear regression of the logarithmic form. The relationship was successfully tested under the operating context of one VM and three VMs. The model is limited to processor-intensive workloads; indeed, this is not surprising, since the predictor is (virtual) processor utilization. However, a conclusion can be drawn: the accuracy of the model in predicting the dynamic power of an individual VM is independent of the concurrent operation of other VMs (albeit for a limited range of workload types).

In [75], it is shown that estimation based on processor utilization can accurately predict the power consumption of multiple concurrent VEs. The prerequisite is that a model for that VE’s workload type has been learnt. The prediction accuracy for individual VEs is independent of concurrent operation. The same conclusion can be drawn for [61]; furthermore, here the range of workload types has expanded to a broader range. With regard to the results obtained in [84], the conclusion is yet again the same: individual prediction accuracy (albeit workload-type-dependent) is independent of concurrent operation.

**Variability in both workload type and number of concurrent, co-hosted VEs** is manifestly claimed in [62,69,81]. It is noteworthy, therefore, that not only is the modeling system adaptable, but it also produces a model that is itself adaptable without the need for real-time adjustment.

We conclude this sub-sub-section with the observation that limited adaptability (defined as independence of workload type and number of concurrent, co-hosted VEs) has been achieved. From the surveyed RUs, we conclude that, presently, the limits of adaptability are the following:Workload type: the power consumption of processor-intensive, memory-intensive, disk-intensive workloads and mixes thereof has been modeled by a single modeling system in an automated manner;Co-hosted, concurrent VEs have been modeled up to but not exceeding over-commitment of processor physical cores.

Caution has been exercised in claiming these limits. For example, while the commitment of logical cores (e.g., Intel Hyper-Threading logical cores) has been investigated, we have found no evidence that power consumption has been modeled in a manner that automatically adapts to a transition of consumption from physical to logical cores.

#### 3.1.3. Lack of Use of Metrics of Energy Efficiency and Standards to Address This Deficiency

We note several experiments, e.g., [55,56,68,71,103,104,105], that target power consumption but less than 12% of the RUs in our corpus approach the problem in terms of some energy efficiency metric [49,50,65,85]. It is necessary to move beyond measurements of how much power was consumed, to measurements of how much power was consumed to carry out a specific task. This change in approach facilitates comparison between research works. More importantly, it directly addresses the question about the cost of operation of infrastructure.

This approach requires identification of a unit of comparison that transcends the boundaries of disparate systems that deliver this unit. This unit of comparison is referred to in the Life Cycle Assessment (LCA) framework (ISO 14040) [106] as the ***functional unit***. A definition specific to telecommunications equipment is given in [107]: the functional unit is defined as “a performance representation of the system under analysis”. This definition is too broad; therefore, units specific to a variety of classes of equipment are defined too [107]. In our corpus, two approaches we have seen are hash/J [50] and J/Web Interaction [65]. The functional units in these cases are the performance of one crypto-hash and one web interaction, respectively. Another is to define a functional unit specific to a digital service delivered over a telecommunications network, e.g., ten minutes’ time of browsing [108]. Note that L.1310 [107] recognizes both metrics where energy is in the numerator [65] as well as those where it is in the denominator [50]. Guidance is available: energy efficiency measurement for several NFV components has been standardized, as well as measurement standards for servers, switches and virtualization systems [109].

#### 3.1.4. Trends

Here, we present trends which we have identified through our collation of problems, approaches and developments. This sub-sub-section is divided into four parts, regarding trends in: challenges, complexity of tackled problems, approaches and developments. Within each part, trends are numbered using Arabic numerals, to differentiate cleanly between them.


**Challenges**


1.The causality DAG (Figure 18) shows that the research space can be characterized succinctly, note the thickness of the links originating at P7 (VE resource use and measurement) and P8 (how to attribute host system power to VEs).
a.Figure 1 and Figure 20 show that the challenge-category tackled most frequently ($R_{P_{7}}=25.3\%$) is how to load the VE’s resources and measure the loading (P7). P7 also has the most frequent presence in RUs: $F_{P_{7}}=65.7\%$) (Figure 20).b.The accurate specification and measurement of load is essential to model formation. These measurements provide the aggregated (i.e., indiscriminate of which VE is consuming) predictors (the predictors are the input side of the model). Disaggregating the predictors and attributing measured power consumption to the individual VEs (the output side of the model) constitutes the second most frequently tackled challenge ($R_{P_{8}}=16.5\%$). P8 also has the second most frequent presence in RUs: $F_{P_{8}}=42.9\%$.c.In the following, the vector (P, A, D, weight) will be used to indicate a path (P, A, D) and the number of triads (weight) through the path:
i.A total of 34.2% of triads collected regard P7 and P8, respectively. The triads graphic (Figure 19) shows that efforts rooted in these two challenges converge on a common objective: building adaptable models of VEs’ power consumption, notably using regressions to linear combinations of computing resource predictors; ii.The set of triads leading to linear regressions consists of {(7,10,5,20), (8,17,5,13), (7,9,5,12), (8,1,5,10), (8,2,5,8), (7,8,5,6), (8,12,5,4) and (7,6,5,3)} and these account for 11.3%.2.The third most frequently tackled challenge is that of the estimation of the virtualization-host-system power consumption ($R_{P_{8}}=15.4\%$). This category might be overlooked in a first inspection of the research space, as it might seem futile to attempt to estimate a power consumption which can be measured. However, in practice, the logistical challenge of the measurement of horizontally scaled system deployments seems to be well known and several works have been undertaken to develop software meters.3.Several significant links originate on P1 (profiling power consumption’s dependency on architecture). The DAG (Figure 18) indicates that approaches to tackling P1 are spread across a mixture of managing resource provisions, the use of synthetic (resource-specific) workloads and predictions using system software’s instrumentation.4.Another large group of links originates on P3 (estimation of host system power consumption). The DAG (Figure 18) indicates that a primary concern in tackling P3 is the type of model to select. The most common choices are linear and non-linear; machine-learning techniques are the least common of the three. The thickness of the triad (3,2,10,11) (D10: relative accuracy of formal approaches) indicates that there is already significant interest in whether the advanced models are worth the effort to develop them and the computational resources required to operate them.5.While $F_{P_{7}}=65.7\%$ (loading VEs and measuring their use of resources), only 8.6% ( $F_{P_{7.2}}=8.6\%$) of all RUs investigate loading containers and measuring their use of resources. With virtual data plane devices, the figure is even lower: $F_{P_{7.3}}=8.6\%$. This imbalance suggests that there is much room left for research into modeling power consumption by containers and data plane devices.


**Challenge complexity**


In [1], we suggest that the diversity of approaches is a metric of the complexity of the challenge. We note (Figure 20) that challenge complexity ($W_{P_{k}}$) generally follows the frequency with which a particular challenge-category is tackled. That is: the more frequently the challenge is addressed, the more diverse are the approaches applied to it. This can be verified by noting that the heights of both sets of bars in the chart (superimposed on the same graphic) follow roughly the same pattern. However, some categories do stand out. For example:

1.P6 regards the influence of temperature on the power model. Approach diversity is poor here because only the use of additional instrumentation can be attributed to this challenge. Model bias might be attributed to this challenge too but, largely, model bias is determined by other challenges within the scope of the RU.2.While host system power attribution (P8) has the second highest research interest (and frequency of occurrence, $F_{P_{k}}$), the number of approaches taken to solve this challenge is relatively small.3.On the other hand, power consumption’s dependencies are tackled by a disproportionately large number of approaches. This is not surprising, as the objects of study (architecture, virtualization platform, virtualization genre) are multi-faceted and dependencies can be investigated through a variety of approaches.


**Approaches**


Our approach utility metric, $U_{A_{k}}$ [1] (Figure 21), seems to be a useful one. It communicates clearly what we have observed in our surveying. Below, we draw attention to saliencies perceived during surveying and confirmed by the metrics.

1.Instrumentation of the consumption of computing resources (A10—which includes microarchitectural instrumentation and that abstracted by system software) is repeatedly adopted ($U_{A_{10}}=15.5\%$, $R_{A_{10}}=15.7\%$) in empirical work in this field. It is also the most utilized of all approaches. In comparison, workload instrumentation accounts for 8.2% utilization ($R_{A_{11}}=5.2\%$).2.Resource-specific workloads (A8) are the more utilized approach to loading VEs (14.3% of all triads). This approach category is the second most utilized. Workloads representative of real use (A9) account for 9.0% of all triads. The corresponding frequency of occurrence figures ($R_{A_{k}}$) are 13.4% and 10.0%, respectively.3.Two other high-utility approaches are (a) modeling bias (A2, with $U_{A_{2}}=11.9\%,\;R_{A_{2}}=11.6\%$) and (b) the managed provision of resources (A6, with $U_{A_{6}}=10.2\%,\;R_{A_{6}}=10.8\%$). The use of representative workloads (A9) follows at $U_{A_{9}}=9\%,\;R_{A_{9}}=10.1\%$.4.A comparison of the patterns of bar height distribution for $U_{A_{k}}$ and $R_{A_{k}}$ reveals that some categories stand out.
a.While A12 (direct power measurement) is not as frequent as the other software-based forms of instrumentation, this approach has a utility that sticks out of the pattern (Figure 21). The reason is that most, if not all, developments obtained in an RU that include this approach depend on the directly measured power.b.Similarly, while workload characterization and instrumentation (A11) are employed with a frequency that is about one-third that of its alternative (i.e., computing resource instrumentation, A10), it has a far better utility-to-frequency ratio than A10. The triads graphic (Figure 19) indicates that one important cause of this high utility is that software-L2-data plane switches and (virtual) network functions are investigated primarily using measurement of workload (and not measurement of computing resources consumed to process a workload).


**Developments**


1.A total of 55.6% of all identified developments are obtained in modeling power consumption. The remaining 44.4% regard how power consumption depends on implementations. The implementations investigated (for their impact on power consumption) range from entire virtualization platforms (e.g., KVM) to components (e.g., processors).2.The most frequent ($R_{D_{12}}=14.5\%$) category of developments is that which regards observations on the dependency of power consumption on architecture. The cause of this high frequency is that developments in the study of architecture establish directional (negative, positive, neutral) correlations rather than predictive forms. For example, in a single RU [67], all of the following developments emerge:
a.D12.08: VM power consumption increases linearly with a vCPU frequency of operation when the vCPU is operating at 100% utilization;b.D12.09: Virtualization-host-system power consumption increases linearly with the number of physical cores operating at 100% utilization;c.D12.1: Virtualization-host-system power consumption increases with the number of VMs active on the same core.3.Amongst developments of models, the most common category (D8) is of the type where power consumption (host or VE) is predicted in terms of workload characteristics ($R_{D_{8}}=12.0\%$). The next most common category regards the prediction of power consumption as a linear function of computing resources ($R_{D_{5}}=8.5\%$)). Machine-learnt models of VE power consumption comprise the second least frequent category, with $R_{D_{7}}=1.7\%$.4.Despite the frequency of developments in category D10 (Figure 22), few works [60,62,63,77,79] compare the accuracy of model types. Models have largely been treated as a means to an end, with little investigation of their relative accuracy and range of validity (in the seven dimensions of variability). This may be a reflection of researchers’ interests. As the popularity of methods from the body of knowledge of data science increases, works (e.g., [63]) that span a broader range of model types may be expected to increase in concert.5.Adaptable, non-linear VE models (D2) occupy a modest 4.3%. Such a distribution adds weight to the observation that this research space is ripe for exploration using advanced modeling techniques. Indeed, one attraction for data scientists is the relative ease with which data can be collected. However, polynomial and other types of regression to closed-form are problematic as suspicion of over-fitting increases with the order of the polynomial. For example, in [76], a sixth-order polynomial is suggested to model the relationship between processor utilization and power consumption by a host.

#### 3.1.5. Three Levels of Abstraction

We note that existing power models may be classified into one of three levels of abstraction. In ascending order of abstraction, these are:Microarchitecture and architecture;Simple characterization of workload;Complex characterization of workload.

The meaning of “abstraction” specific to our use here is perhaps most easily grasped by referring to the variables used as model inputs. In all cases, the variables are some measure of load. At the lowest level, inputs that quantify the operation of processor sub-units are used (event counters and event timers). The highest level uses inputs that quantify the demand for a telecommunications system or service. Clearly, the latter inputs are far more detached from the underlying, concrete implementation than the former.

With regard to the lowest level of abstraction, many power models use hardware resource consumption to estimate the power consumed by virtual components. One survey [105] (while comparing the available power models for processors, VMs and servers) observes that most of the power models for virtual machines use physical machine counters to estimate the corresponding resource utilization by the virtual components. We offer further insight on this matter. We concur in the observation that much current research is concerned with modeling power consumption of virtual machines. The approach may be succinctly described as estimates obtained from models trained out of either architectural or microarchitectural instrumentation data. Note that we distinguish between architecture and microarchitecture using the classical interpretation [110]. Now, the term “architecture” is severely overloaded and its interpretation can easily differ from that which we wish to exploit. In the following description of the levels of abstraction, we use the alternative “system software’s instrumentation” to convey the same meaning as “architectural instrumentation” with less ambiguity.

**Microarchitectural instrumentation** is the lowest level of instrumentation. Power consumption is expressed in terms of variables that are defined at sub-CPU and sub-subsystem levels. The granularity of this level holds the greatest potential for accuracy, but the rate of change of the observed variables has deterred several researchers from pursuing this approach to instrumentation, citing concerns about communicational and computational overhead. This concern has been dismissed by three groups of researchers [70,86,98] who have indicated that acceptable accuracy can be achieved with negligible overheads. Given the proliferation of works based on this approach and the availability of high-level (programming) language code (HLL) that facilitates the use and the potential for the capture of physical behaviors, then ***our general recommendation is a preference to investigate the use of microarchitecture instrumentation***.

**System software’s instrumentation** regards a class of instrumentation that has meaning across the spectrum of computer systems. The input variables, such as the CPU or network interface utilization, are produced by some digest (function) of intermediary system software. For example, an operating system’s (OS) measurement of core utilization can roughly be described as the core’s duty cycle on behalf of the OS. This is significantly removed from knowledge of activity within the core. While models at this level are more abstract, they are still low-level, especially when compared with the other levels.

**Simple characterization of workload** is a less frequently encountered abstraction, used by Enokido, Takizawa and various others with whom they have co-published [68,71,111]. These models describe power consumption in terms of fundamental descriptors of workload, e.g., the number of processes and transmit/receive data rate.

**Complex characterization of workload** is the least granular of the models in the survey [34,112]. The objective here is to quickly proceed to a good estimate of the power or energy required to produce the workload. This kind of model has no use in real-time control ***but it is useful for macroscopic comparisons***, i.e., comparisons between two disparate systems for the provision of a service. The comparison might regard two different paradigms of provision of the service, e.g., classical vs virtualized implementations. Thus in [34], the implied (system) metric is the amount of power required to deliver 1 million packets per second of throughput through an evolved packet-core’s (EPC) serving gateway (SGW). In [112,113,114], the objective is to minimize the amount of consumed power by virtualizing baseband processing functions, evolved packet core, customer premises equipment and radio access network functions.

#### 3.1.6. Service Determinism: A Criterion Particular to the Telco Cloud

The NFV data plane demands determinism [37,96,115]. Strictly, ***service determinism*** is sought since the packet arrival process is generally unconstrained. The root cause of this need is to correctly size equipment resources to meet load demands, whether throughput, latency or jitter. ***This need is intense***, as it impinges on a PTNO’s obligations, specified in legally binding service-level agreements (SLAs). Service determinism has been approached through the augmentation of GP hardware, with domain-specific architectures (DSAs, which we first referred to while describing problem category P1). Hardware-centric DSAs for the NFV data plane are constructed out of elements (or systems thereof) that can be divided into three groups.

**IO system architecture:** This group comprises the abstraction, ***directly at a peripheral interface***, of functionality that facilitates the virtualization of hardware, e.g., SR-IOV (used in PCI—Peripheral Component Interconnect) and N-port ID virtualization (Fibre Channel).**Processor architecture:** This comprises architectural change that facilitates the partitioning of processor resources, such as multi-core processors and NUMA.**Co-processing:** Compression and decompression, encryption and decryption and packet header processing are examples of high-volume tasks that can be offloaded to co-processing subsystems. Examples include Intel QuickAssist Technology (QAT) and TCP Offload Engine (TOE).

We further observe a set of approaches that are complementary to DSAs in the quest for determinism. These include:**Large memory pages:** exploitation of the facility to organize virtual memory into larger pages than the general-purpose 4 KiB;**User space programming:** diversion of the control of hardware resources away from the multi-service kernel, to single-common-use user space programs, e.g., DPDK and AF_XDP.

Notably, resource partitioning (see processor architecture, above, i.e., the second hardware-centric approach), combined with user space programming (the second complementary approach), realize the run-to-completion scheduling model [96].

We conclude with an observation on ***the cost of the current realizations of service determinism***. The (aforementioned) combination of core partitioning and user space programming has been widely adopted through Intel’s popularization of DPDK, via Intel’s open-source liaison efforts. The multi-core processor enables service determinism through an approach that is anathema to the principles of multiprogramming: the dedication of hardware to a specific task.

#### 3.1.7. Direct or Indirect Measurement of Power in Virtualized Environments?

We observe that most research in modeling power consumption seeks to obviate the need for direct measurement through indirect measurement. This indirection consists of a measurement of resource use which has a discoverable relationship with power consumption by the entity hosting the resources. Modeling, here, has the objective of indirect measurement of a variable that is not directly accessible (power consumption by VEs), through others which have convenient and reliable instrumentation. The accessible variables are referred to as ***power proxies***. The RAPL interface provides a unique approach to measurement as it directly addresses power consumption. However, notwithstanding appearances of direct measurement, RAPL is actually based on a software model that uses performance-monitoring counters (PMCs) as predictor variables to measure power consumption. It is available in processors starting from the Sandy Bridge microarchitecture. RAPL measures the power consumption of different physical domains, where each domain consists of either cores, sockets, caches or GPU. We briefly comment on its accuracy through references to research that has investigated them.

In [98] the advantages and drawbacks of using RAPL were investigated. Different Intel architectures such as Sandy Bridge, Haswell and Skylake were used in the experiments to analyze RAPL’s accuracy and overhead. Data collected were modeled using a linear model and a Generalized Additive Model (GAM). The accuracy of the predicted results was compared with the measured power consumption from a precise external hardware power meter where RAPL based models show 1.8–4.3% of error for the various architectures. The prediction accuracy of RAPL-based power models was also compared with those based on OS counters, where OS-based models show a higher error of 5–16%. Furthermore, the performance overhead (in terms of time) of using RAPL was studied at different sampling frequencies and for different application runs. Results show that even with a high sampling frequency of 1100 Hz, RAPL incurs overhead of not more than 2%. Some limitations of using RAPL include poor driver support to read energy counters, the overflow of registers due to their 32-bit size and the measurement of energy consumed by individual cores.Another study to analyze the precision of RAPL is presented in [116], where only the dynamic change in power consumption is observed. An external power measurement unit (WattsUp Pro) is used as a reference for power measurement values. Intel Haswell and Skylake servers were used in the experiments to run different applications and to find the reliability of RAPL with the help of external power meters. However, in this research work, only two power domain packages (power consumption of whole socket and DRAM domain of RAPL) were observed. Applications such as dense matrix multiplication and 2D Fast Fourier Transform were used for server power profiling. Results show that the power measurement error varies with changing applications and its workload size. For different applications the average measurement error using RAPL was in the range of 13–73% considering WattsUp power meter as the ground truth. It was concluded that with the modern multi-core parallel processing and resource contention for shared resources, there is a complex non-linear relation between performance, workload size and energy consumption. Hence it is difficult to attain low error percentage for power measurement using on-chip sensors.

### 3.2. Research Gaps

Three significant challenges remain unaddressed, while a fourth requires further attention:The modeling of containers’ power consumption;The effect of overcommitment on power efficiency;The investigation and classification of DPDK applications;The fourth challenge, which is starting to receive some attention [83,85,87,94], regards the modeling of power consumption by virtualized I/O.

Note that our treatment of research gaps does not address improvements in approaches. As we have already indicated, it is in our treatment of developments that more adaptable modeling methods are required to tackle the dimensionality of the field. Similarly, we do not include the lack of use of metrics of energy efficiency with research gaps, as it is a deficiency in the approaches, not a challenge in itself. Rather, here our attention focuses on where the more pressing challenges lie for the development of power and energy control of VEs.

**Gap #1: Modeling of containers’ power consumption.** Few works [78,81,82] tackle containers from the perspective of their power consumption. However, at least for the telco cloud, VMs are no longer the destination (see, for example, [40,117]. Containers have replaced virtual machines as the base for deployment of virtual network functions. In [78], the approach(-set) taken is to:Use representative workloads, e.g., HEVC (High Efficiency Video Coding) transcoding and machine learning image classification;Customize the set of low-level instruments used to correlate power and energy consumption with workload characteristic.

This work develops a profiling tool. It provides guidance that is specific to the application and both the hardware and software aspects of the containing platform. “[P]olicy” for “tradeoff between energy, power and application performance” is the cited objective. Given the high-dimensionality of the core challenge, this set of approaches to the modeling and measurement of power consumption, may well be more coherent with the European Telecommunications Standards Institute’s (ETSI) Management and Orchestration (MANO) standard. Such information would then be included in the infrastructure-resource-requirement’s meta-data descriptors in the VNF package [118].

**Gap #2: The effect of overcommitment on power efficiency.** Overcommitment consists of the allocation of more capacity of some compute resource to VEs, than is physically installed. The concept is very similar to the oversubscription of telecommunications capacity to subscribers, such as when the arithmetic sum of capacities of access links exceeds the aggregating device’s backhauling capacity to a central office/local exchange. In this context, overcommitment principally concerns processing cycles and memory space. As with oversubscription, there is an optimization problem to solve. One problem of interest to this survey’s scope is understanding the relationship (say, ratio) of committed virtual resources to installed physical resources that optimizes the total cost of ownership (TCO) of cloud infrastructure:On the one hand, the facility to overcommit has a direct impact on the density of the packing of VEs (number of concurrently active VEs) on a virtualization-host-system, thereby reducing the TCO;On the other hand, overcommitment may reduce the power efficiency of a workload.

Had this challenge been tackled in any depth, or at least in any breadth, it would have merited a category of its own. Currently, however, we are only aware of a single study [65] that tackled this challenge. The results obtained strongly justify overcommitment of processor cores to vCPUs, for the case of transaction web service workloads, with the increase in throughput (measured in web interactions per second, or WIPS) increasing at a faster rate than power consumption. This behavior was observed well into overcommitment ratios of processor cores to vCPUs equal to 3 (three). The overcommitment of physical to VM memory was not investigated.

**Gap #3: Investigation and classification of DPDK applications.** In Section 3.3.2, we address the relationship between DPDK and power consumption. In the course of a public discussion in the forum offered by the North American Network Operators’ Group (NANOG) [119], there emerged a need for clarity on DPDK’s association with inefficiency in power consumption. Interest was particularly expressed in the knowledge of a classification of extant DPDK applications according to their power consumption, and contribution to the code base to improve power-hungry applications. In summary, among high-performance packet I/O frameworks, a classification of applications that use DPDK APIs in order to assess their power consumption and correct usage would be of particular interest, given their diffusion and potential.

**Gap #4: Modeling of power consumption by virtualized I/O.** Power consumption of network I/O has been investigated to some extent as this is central to the feasibility of network functions decoupled from hardware. While software and hardware solutions are already available (see Section 3.3.2), they require frequency and idling control targeted to their specific operating conditions. Notably, naive DPDK runs the processor core at its maximum power consumption, regardless of load. The exploitation of adaptive-rate (AR) processing and low-power idle (LPI) should provide a means to save power while processing high networking loads. However, effective control of these means is still elusive, despite both using Xeon Haswell microarchitectures, refs. [83,87] reach opposite conclusions about the feasibility of processor core C-states. The former [83] finds LPI an effective means of reducing power consumption of packet forwarding (with limitation on latency) while the latter [87] finds it ineffective, preferring the use of the **pause** instruction. Furthermore, in [120], performance state transitions (P-state) are found to impose a high transition latency, while in [87], P-state regulation is the preferred approach. There is scope for research in the dynamic adaptation of the processor’s operating state to save power.

### 3.3. Pitfalls

#### 3.3.1. Power Consumption Does Not in General Increase Linearly with Processor Utilization

Notwithstanding advances made in identifying operating contexts that manifest a sub-linear power-utilization relationship [62,69], more recent publications [57,121,122,123,124,125,126,127,128] persist in using the linear model without acknowledging its limitations. The model is simple to use and has some foundations in research [129]. It has three premises, described here with regard to the operation of Microsoft Windows:When Windows has no threads to run on a logical core, it schedules the idle thread [130].The idle thread keeps the processor in a low-power state [131]. The specific state depends on the processor’s green capabilities.In the complement (non-idle time), the processor issues instructions at a constant rate.

This simple model has limitations [62,132]. It fails to take into account diverse processor operating contexts, some of which are coming to bear on current use cases. Specifically, the third premise is true only to the extent to which instructions are being fetched and data are being loaded from/stored to an instruction and data cache, respectively. Consider the context of 90% and greater hit ratios. At such cache hit ratios, the rate of instruction issue is expected to be narrowly distributed about its mean. By contrast, the lower the hit ratio at the cache level before main memory, the lower the fraction of non-idle time at which power consumption saturates. This saturation is strikingly illustrated in [62]. The variation of power consumption due to the execution of tests from the SPEC CPU2000 benchmark suite is shown. The power consumption diverges at 25% CPU utilization and the consumption of the processor-bound test (mesa) is greater than that of the memory-bound test (mcf) ***by a factor of about 2.6***. 

Another good (albeit broad) illustration of this pitfall is given in [133]. Data on power consumption and CPU utilization under a standardized benchmark are plotted for four different physical server models. None of the relationships are linear. Neither is there a single, common behavior.

Researchers align themselves into two groups with regard to CPU utilization. One group favors (operating-)system metrics (of which CPU utilization is one metric) and the other favors event counters (microarchitectural instrumentation). The arguments posed by each group against the other’s approach can be summarized as follows. The “system metrics” group claims that the “event counters” group’s work is (a) not portable (at least across microarchitecture families) and (b) cannot be exercised without low-level access to the host (therefore, this approach cannot be exploited by user-level privileges) (see, e.g., [134] and [75]). The “event counters” group claims that CPU utilization is a workload-dependent predictor (see, e.g., [59]) and therefore cannot be used without re-training the model. Indeed, this modification to the “system-counters” approach is employed in [76], where it is stated that “[b]ecause of changes of VM’s internal applications…parameters must [be] recalculated automatically”. ***Given these arguments, it seems that the system metrics group argument is weak: both system metrics and event counters require re-training if hardware is changed but system metrics lack the granularity to discriminate between workloads*** (cf. [62]). This means that CPU utilization can only be used as the sole predictor if it is re-trained with a change in workload. We deal with this problem, which we have termed the fallacy of the universal model, in Section 3.4.1. 

With regards to the use of hardware threads (Intel® Hyper-Threading), we have observed that various works concur on the operating context under which a linear relationship is subject to the lowest error. This includes at least the following two conditions:The processor cores are increasing their instruction issue rate in proportion to the fraction of time they spend busy. This implies that instruction and data cache hit ratios are high. This is simply the third premise;Only one logical core is active per physical core at any given time [62,63,69]. Expressed alternatively, actual utilization must lie below half maximum utilization. The underlying cause is that activation of the second logical core employs fewer organizational units of the processor than activation of the first logical core.

The first condition is particularly problematic, as cache miss ratios are likely to be much higher in the context of virtualized environments. In such environments, the number of runnable threads is the sum of runnable threads controlled by independent operating systems. Evidently, this is higher than the expected number of runnable threads on a single server instance.

Other evidence of this “utilization trap” is not hard to find. In [57], the compute resource is stressed using cpulimit and stress-ng. The “cpulimit” utility runs a specified process image, then pauses and resumes it until a certain percentage utilization is reached [127]. The repetitive execution of a single process is highly likely to create conditions for very high instruction- and data-cache hit ratios. Such favorable hit ratios skew results towards the linear relationship between CPU utilization and power consumption.

#### 3.3.2. DPDK Is Not Intrinsically Inefficient in Power Consumption

Research on power efficiency in DPDK applications [48,51,53,87] has portrayed DPDK as power inefficient. Before proceeding to our exposition of this pitfall, we distinguish between data, control and management planes. “Data plane” is a term used to refer to the infrastructural means that provide the capacity for exchange of customer (or subscriber, or end-user) data. It is complemented by a control plane, which refers to those means that facilitate the dynamic setup, maintenance and tear-down of a functional data plane. Another complementary part is the management plane. This includes the infrastructural means for a network operator to configure and monitor the control plane and the data plane, as well as intervene to correct faults arising in either plane. Simpler networks may have no control plane. 

Now, we proceed to the exposition of the pitfall. In one particular case [48], it is claimed that “we found that a poll mode driver (PMD) thread accounted for approximately 99.7 percent CPU occupancy (a full core utilization).” The implication that seems to emerge here is that the PMD itself is driving this power consumption.

This portrayal is problematic at best and incorrect at worst. The referenced investigations of DPDK have indicated a very low power efficiency, **but they do not clearly distinguish between the responsibility of the DPDK API and the application using it** (the API). A recent, public thread [119] has emphasized the responsibility of the application developer in the avoidance of the naïve, “default approach” of busy polling. Such an approach would, indeed, poll network IO hardware continuously [48], truly fitting the epithet “spinning-hot” [37]. However, a broader (in the sense of including industrial correspondents) investigation [119] suggests that:Contrary to claims in [48], it is the driving behavior of OvS that is inefficient in power consumption;There are simpler, technical means of throttling a polling loop, including, say, the use of program code to interleave ACPI C1 states with polls according to traffic demands.

These latter observations cast doubt on the claim that automated frequency control is outside the scope of current frequency governors, since “the OS won’t be able to distinguish whether it’s under a heavy load” [48]. On the other hand, savings through NUMA awareness [51] (where transmit/receive port, memory and processor core are kept within the same NUMA node) is affirmed in [119].

#### 3.3.3. Research on Power Models without Power-Relevant Context

This pitfall traps readers who attempt to draw conclusions from published research which lacks a clear specification of context relevant to power consumption. The pitfall is best illustrated through examples.

**Failure to emphasize context: idle power consumption vs frequency**. The dependence of idle power consumption on clock frequency is context sensitive. In [49], it is explicitly stated “idle power consumption remains constant, regardless of the CPU frequency … across the whole frequency range” (1.6–2.6 GHz). The CPU is an Intel Xeon E5620. In [81,82], a quadratic relationship between idle power consumption and frequency is observed. Here, the CPU is an Intel Core i5 Haswell. In these two instances, emphasizing the restricted scope of findings would suffice to spare a reader from excessively broad inferences.**Failure to emphasize context: idle power consumption vs hardware and software specification**. Enokido’s and Takizawa’s work [71] derives a power consumption model for a server while VMs run computation-bound processes. The servers used run on Intel Core i5-3230M processors. These processors are used in the mobile device market [135]. They are capable of low-power idle states [136]. CentOS 6.5 uses a tickless kernel [137]. Combined, these facts, relevant to the context of power consumption, provide a plausible explanation for the observed increment in power (denoted, in [71], by $\min C_{t}$), when a core in a package is activated. Again, therefore, the scope of findings is likely to be restricted.**Failure to fully define context: Configuration of power-relevant parameters.** We use [138] as an example. No reference is made to whether Hyper-Threading is enabled. This is essential to understanding how the ESXi vCPUs are created. Neither is any information given about how the vCPUs are related to physical (or logical) cores. Nor are we told how virtual network interfaces and switching are set up. ESXi version 5 offers both paravirtualization (“vmxnet”) and emulation (“e1000”) to implement virtual network interfaces. The impact on energy consumption of selecting a virtual network interface implemented by emulation can be expected to be high [56].

The examples cited illustrate the importance for a researcher of power models to qualify his/her results ***with a well-defined physical context***. Research into power models involves hard components and a diligent characterization thereof is essential to the acceptance of work as scientific research.

#### 3.3.4. Benchmarks May Skew Power Consumption According to Their Organizational Dependencies

We have seen that both “cpulimit” and “stress-ng” do not produce generally representative measurements of power consumption. This observation is not limited to the measurement of power consumption. The use of kernels, toy programs and synthetic benchmarks to measure performance has been identified as unrepresentative [110] of general performance. Benchmarks are standardized workload generators that are used for the comparison of computer systems for a specific class of application. Unless this application class is a good representative of the application of the computer system in productive use, the power consumption measured under test is not a reliable predictor of that obtained during productive use. It is necessary to plan test workload generators in advance and state the limits of the validity of results. In [65], TPC-W is used, which is a transactional web benchmark that can simulate the business-oriented online web servers. The MySQL++ Java version of TPC-W benchmark, suitable for cloud applications, is used to generate the online traffic, where three different traffic profiles based on the browsing, purchasing and ordering of books are generated. The throughput measure for these servers is observed through the metric Web Interactions Per Second (WIPS).

#### 3.3.5. Processor Organization Significantly Impacts Power Consumption

We illustrate this point with a wide-ranging example [139] which compares the Intel Xeon X5670 and AMD Opteron 2435.

1.Different idle loops (using no operation, pause, repetition, etc.) were tested to see their effect on power consumption of both systems. It was observed that the Intel Xeon has a loop stream detector, which disables the processor’s features such as fetch and decode. On the other hand, the AMD processor has no hint to process these loops efficiently; hence, it consumed more power than the Intel processor.2.A processor consumes a different amount of power depending upon the instruction (such as load, addition, multiplication, etc.) and the level in the memory hierarchy which is accessed by the instruction.
a.For the Xeon, data throughput of all instructions from a particular memory hierarchy level is almost the same, but there is a difference in their power consumption. The ‘load’ operation consumes the lowest power compared with other instructions, and this holds true for all memory hierarchy levels. The reason is that the ‘load’ instruction just needs to load the content on the processor registers whereas ‘add’ and ‘mul’ operations are more computationally demanding.b.However, the AMD processor’s behavior is the opposite. When the ‘load’ operation accesses the L1 cache, it achieves almost one-and-a-half times the data throughput of other operations and hence also consumes more power. This difference in resource utilization is due to the different microarchitecture of AMD processors, where the ‘load’ instruction is handled by many floating-point pipelines. Other instructions just use a single pipeline for their operations. Moreover, AMD processors have an exclusive cache level design, which requires write-back when evicting data among different cache levels. Since Intel’s inclusive cache design does not require this function, it consumes less power. Within higher memory hierarchy levels (L2 or L3 or main memory), the AMD’s computation (‘add’ and ‘mul’) and data transfer operations (‘load’) deliver roughly the same data throughput and consume roughly the same power.

#### 3.3.6. Isolation of VE for Power Modeling and Measurement

Isolation of any VE from its hardware counterparts cannot be achieved completely [140]; thus, the assumption of measuring power consumption of an individual virtual entity irrespective of the hardware on which it is implemented is an illusion. The virtual infrastructure is composed of several components at both hardware and software level, where the effect of underlying hardware, OS and VNF technology can significantly impact the power consumption. Hence, isolation as well as the modeling of power consumption for an individual virtual component is difficult to obtain.

### 3.4. Fallacies

#### 3.4.1. A Universal Power Model

We have suggested that the core challenge in modeling power consumption by VEs lies in the number of dimensions of variability. This has been demonstrated throughout this survey, where a number of generalizations have been addressed. In summary, the literature shows that:Host power consumption does not generally have a linear relationship with processor utilization;CPU-intensive workloads that repeatedly execute the same code skew power consumption results;Network-intensive workloads are power- and time-consuming because they employ emulations of network switches, but the root cause (emulation in the hypervisor software switch) disappears with SR-IOV [141];Host saturation must be taken into account in predicting VEs’ power consumption;Processor utilization (an architectural attribute) is insufficient to predict host power consumption and microarchitectural attributes, such as LLC misses, are necessary to predict host power consumption even for the same level of processor utilization.

This list, while not exhaustive, amply illustrates that the several dimensions of variability are significant in the determination of VE power consumption. A model claiming to determine power consumption as a function of fewer variables than the dimensions we have pointed out ***must be accompanied by a scoping region that limits its use***. While a precise scope may be an unrealistic demand, it is essential that guidance be given about the conditions of the use of the model. We now illustrate this point by using two examples from the corpus.

**Example #1:** Khan [50] compares energy efficiency (hash/J) obtained by scheduling process threads on additional cores, with that obtained by scheduling them on hardware threads on active cores (through Intel Hyper-Threading). He shows that the former is greater than the latter. In apparent contrast, Enokido and Takizawa [68] show that for a given data transmission rate through the uplink of a software virtual switch, greater energy efficiency (W/bps) is obtained by operating an additional hardware thread on an active core (through Intel Hyper-Threading), than operating an otherwise idle core. An important difference lies in the task’s processing “intensity”, i.e., the rate of the supply of instructions. While Khan’s operations are tightly bound to the processor (cryptographic hashing), Enokido’s and Takizawa’s operations are distributed over the processor and network input/output. Without delving into detail, it is realistic to hypothesize that the average instructions per second demanded are far lower in the networking application, since the transmission of a large file (as is the case here) does not take place in one processing burst. The operating time is divided between the processor and the media channel. In such a scenario, the added capacity of the same-core hardware thread suffices.

**Example #2:** At the time of writing, the scope of validity (where the scope is a sub-space of the seven-dimensional space) is typically only implicit. Notably, in [75], a “refined model” is used as a means of the accurate prediction of power consumption by virtual machines while running very specific benchmarks. It is also noteworthy that the authors contemplate a type of onboarding process wherein “new” VM entrants to a cloud are modelled as a prerequisite to their inclusion in the power-prediction system. Indeed, such a process is already intrinsic to the management and orchestration of virtual network functions. Just as the virtual deployment unit (VDU) nodes (in virtual network function descriptors (VNFDs)) store VM properties describing computer system resource demands, so can the descriptor template be extended to provide properties regarding power consumption demands. This “onboarding” is necessary since the selected predictors and modeling do not cover a sufficiently broad range of workload types, and a specific model must be learnt online, i.e., on the fly.

On the other hand, we propose that a comprehensive power model for existing implementations may be possible, under two conditions.

1.Every resource that consumes power must own a counter that registers its usage, or lack thereof, during a specific clock cycle.2.Usage of a specific resource during a specific clock cycle must consume a constant amount of energy. This has the following corollaries:
a.Energy consumption by the specific resource is a linear function of the number of clock cycles for which the resource is active;b.Power consumption of a system can be expressed as a linear combination of the total set of such resources;c.The amount of energy consumption by a specific resource during a specific clock cycle must be independent of usage of other resources during any other clock cycle.

## 4. Conclusions

We conclude by summarizing our contributions (Section 4.1) and suggest a framework for future research into real-time, predictive models of power consumption by VEs (Section 4.2).

### 4.1. Contributions

We have identified seven dimensions of variability (workload type; virtualization agent; host resources and architecture; temperature; power attribution; co-hosted, concurrent VEs; and (clock) frequency of operation) and observed that the challenges tackled have aligned themselves with these dimensions. This breadth has prompted us to emphasize the fallacy of the universal power model: no single power model can cover all seven dimensions through the inclusion of variables and parameters. It is essential that prospective users of any such power model be aware of the limits of its scope. On the other hand, we have pointed out that the state-of-the-art includes adaptable modeling systems that handle variability in more than one dimension. Moreover, at least limited variability in two of the seven dimensions—workload type and concurrent operation of (multiple) VEs—is commonly validated, i.e., whether the model is truly capable of predicting power consumption under variability in workload type and the number of concurrent VEs.

PAD elicits trends in its proceedings through a sample of a corpus. In particular, the following examples are among the most noteworthy (but not the only) saliencies.

The challenge category tackled most frequently is that of how to load the VE’s resources and how to quantify and measure the load; disaggregating the predictors and attributing measured power consumption to the individual VEs is the second most frequently tackled.The variety of approaches that tackle a (category of) challenge is positively correlated to the frequency with which it is tackled.Instrumentation of computing resources (e.g., instrumentation of microarchitectural artifacts) is the most commonly adopted approach (towards developments), surpassing instrumentation of the workload.Resource-specific workloads (e.g., processor-specific) are the most commonly utilized, surpassing workloads representative of real use (e.g., web applications).In developments, the most commonly developed model type is that where the power consumption (of the host or VE) is predicted in terms of workload characteristics; power consumption as a linear function of computing resources is second.At the other end of the frequency range of developments, machine-learnt models comprise the second least frequent category of developed models, and adaptable, non-linear VE models are also very infrequent.

The process of parsing works and aggregating their codes is, however, only the principal ingredient in the overall progression towards the end goal: a set of themes that suitably profile the works in an area of research. Indeed, these codes and their inter-relationships have elicited several research gaps, pitfalls and a fallacy, as well as evidence of the state-of-the-art and of research domains.

### 4.2. A Framework for Development of Real-Time, Predictive Power Models

Evolution of the research space on power consumption in virtualized environments now suggests the following framework for the further development of power models:1.Division of the problem into:
a.A modeling concern:
i.What components to include;ii.What workload(s) to consider;iii.What state factors (temperature, frequency, performance and idle states) to account for.b.An attribution concern, i.e., how to attribute host power to VEs.2.Division of the approach into:
a.Microarchitectural instrumentation, based on intimate knowledge of the microarchitecture and the memory system;b.Granular attribution based on time-division multiplexing;c.Model selection.3.Development of parameterized models, subject to continuing (if not continuous) optimization of the parameters under machine learning.

## Figures and Tables

**Figure 1 sensors-23-00255-f001:**
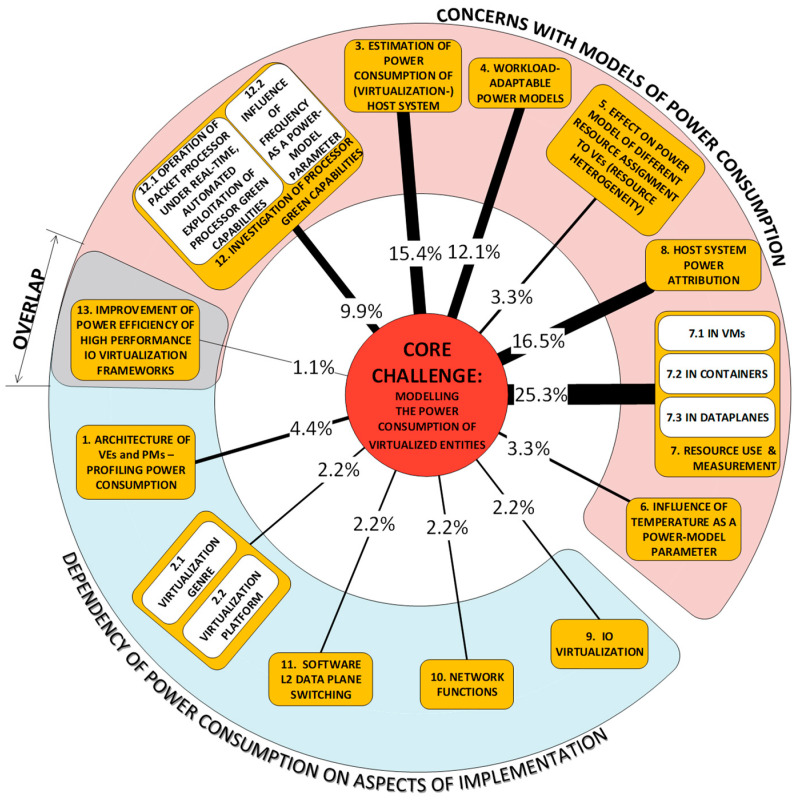
The core challenge and its derivatives; research interest $R_{P_{k}}$ is shown in line thickness and as percentage.

**Figure 2 sensors-23-00255-f002:**
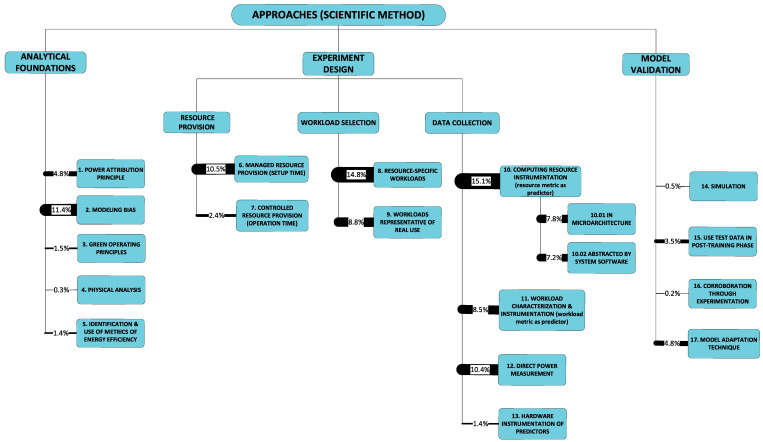
A taxonomy of approaches; approach utility $U_{A_{k}}$ is shown in line thickness and as percentage.

**Figure 3 sensors-23-00255-f003:**
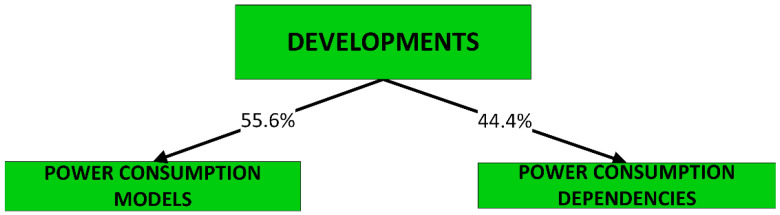
A top-level division of the developments, with frequency of occurrence shown as a percentage.

**Figure 4 sensors-23-00255-f004:**
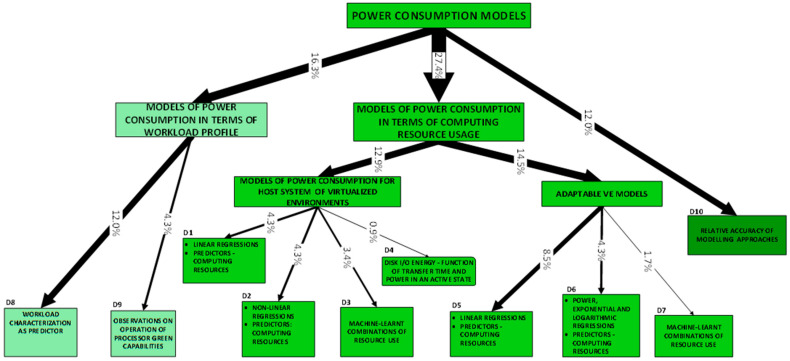
A taxonomy of power models, with frequency of occurrence shown in line thickness and as percentage.

**Figure 5 sensors-23-00255-f005:**
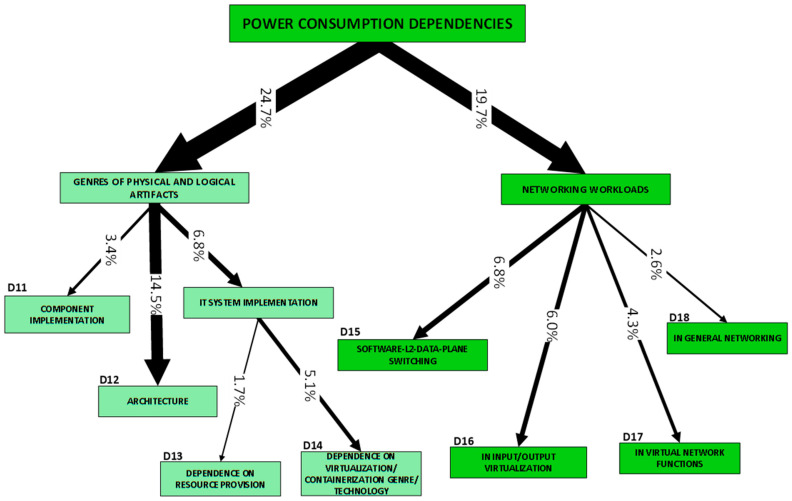
A taxonomy of power dependencies, with frequency of occurrence shown in line thickness and as a percentage.

**Figure 6 sensors-23-00255-f006:**
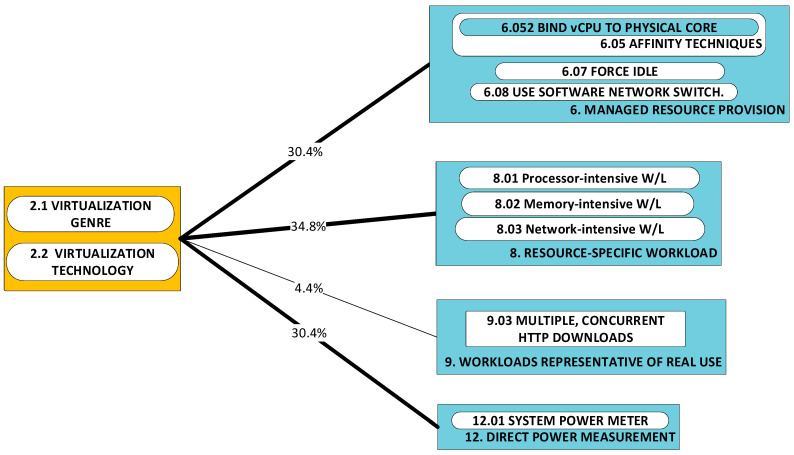
Approaches to solving challenges in category P2; utility metric $U_{A_{k}}$ is shown in line thickness and as a percentage. RUs in the respective categories are the following: P2: [55,56]; A6: [55,56]; A8: [55,56]; A9: [55,56]; A12: [55,56].

**Figure 7 sensors-23-00255-f007:**
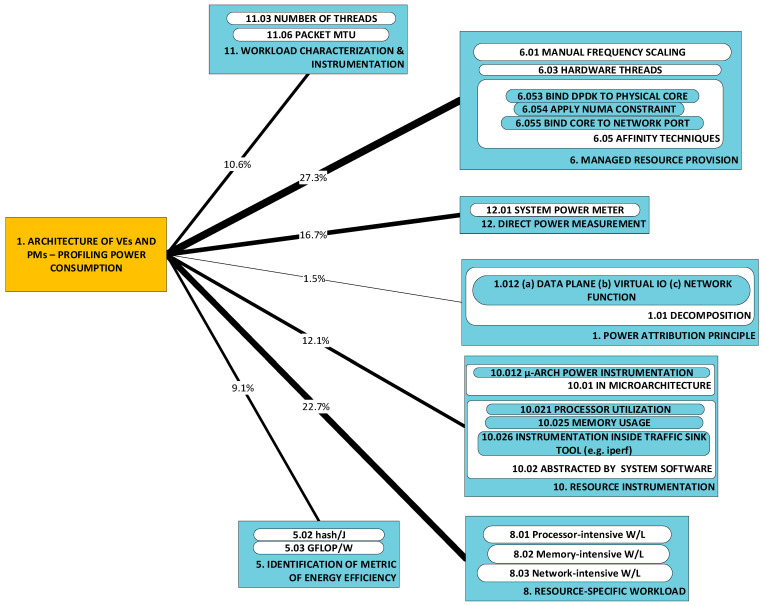
Approaches to solving challenges in category P1; utility metric $U_{A_{k}}$ is shown in line thickness and as a percentage. RUs in the respective categories are the following: P1: [48,49,50,51]; A1: [48]; A5: [49,50]; A6: [48,49,50,51]; A8: [48,49,50,51]; A10: [48,49]; A11: [48,49]; A12: [48,49,50].

**Figure 8 sensors-23-00255-f008:**
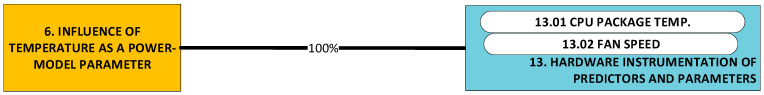
Approaches to solving challenges in category P6; utility metric $U_{A_{k}}$ shown in line thickness and as a percentage. RUs in the respective categories are the following: P6: [60,81,82]; A13: [60,81,82].

**Figure 9 sensors-23-00255-f009:**
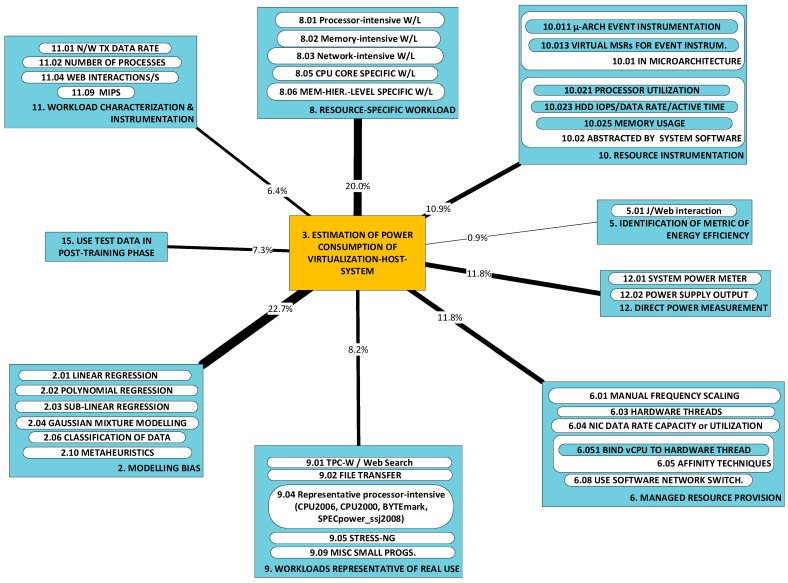
Approaches to solving challenges in category P3; utility metric $U_{A_{k}}$ is shown in line thickness and as a percentage. RUs in the respective categories are the following: P3 [57,58,59,60,61,62,63,64,65,66,67,68,69,71]; A2 [57,58,59,60,61,62,63,64]; A5 [65]; A6 [60,66,67,68]; A8 [59,60,61,63,64,66,67,68,71]; A9 [57,58,60,62,65,68]; A10 [57,58,59,61,62,63,64,69]; A11 [64,65,67,68,71]; A12 [57,58,59,61,62,63,64,67,68,71]; A15 [57,62,63,64].

**Figure 10 sensors-23-00255-f010:**
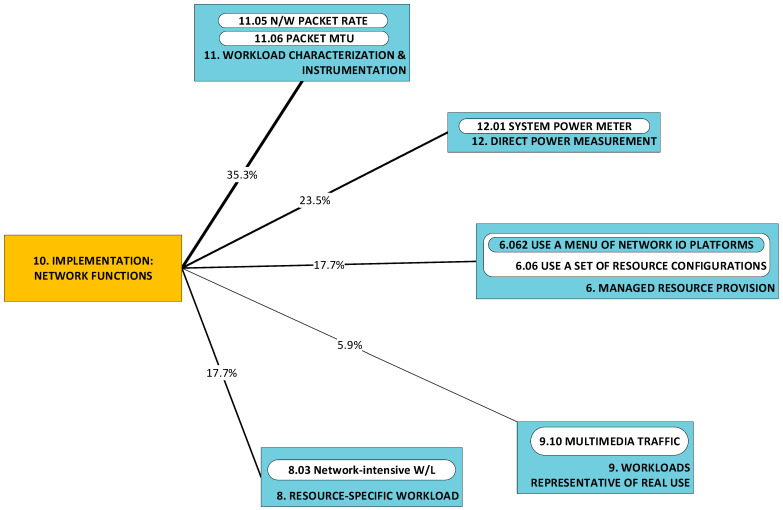
Approaches to solving challenges in category P10; utility metric $U_{A_{k}}$ is shown in line thickness and as a percentage. RUs in the respective categories are the following: P10: [48,53]; A6 [53]; A8 [53]; A9 [48]; A11 [53]; A12 [48,53].

**Figure 11 sensors-23-00255-f011:**
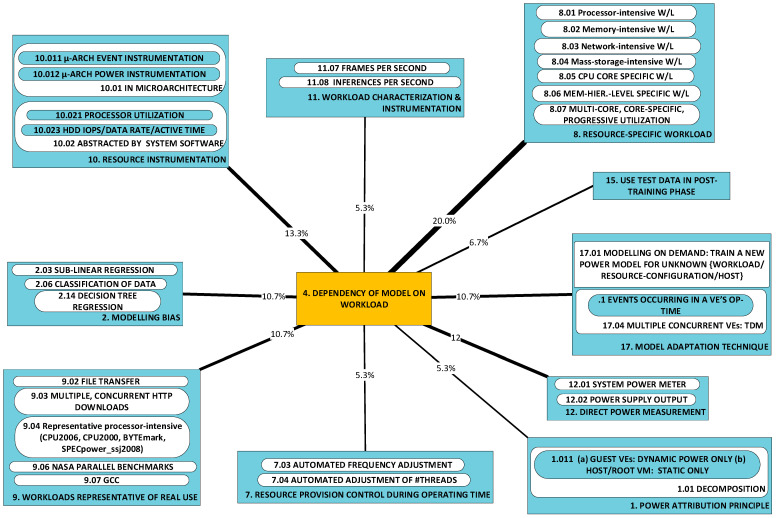
Approaches to solving challenges in category P4; utility metric $U_{A_{k}}$ is shown in line thickness and as a percentage. RUs in the respective categories are the following: P4 [56,62,67,68,69,71,75,76,77,78,79]; A1 [62,77]; A2 [62,77]; A7 [78]; A8 [56,67,69,71,75,76]; A9 [56,62,68,75,79]; A10 [62,78,79]; A11 [78]; A12 [56,62,76,79]; A15 [62,77]; A17 [62,67,68,71,75,76,78].

**Figure 12 sensors-23-00255-f012:**
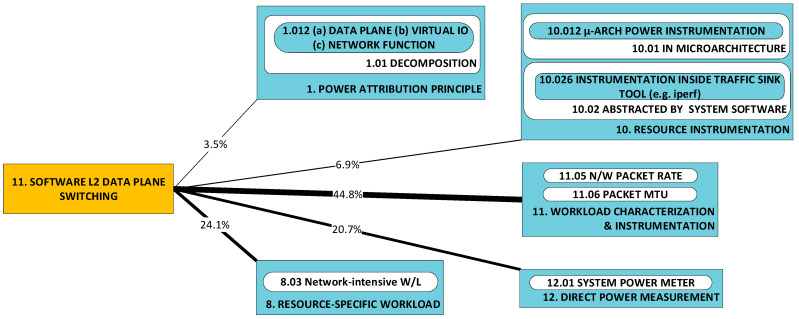
Approaches to solving challenges in category P11; utility metric $U_{A_{k}}$ is shown in line thickness and as a percentage. RUs in the respective categories are the following: P11 [48,53]; A1 [48]; A8 [48,53]; A10 [48]; A11 [48,53]; A12 [48,53].

**Figure 13 sensors-23-00255-f013:**
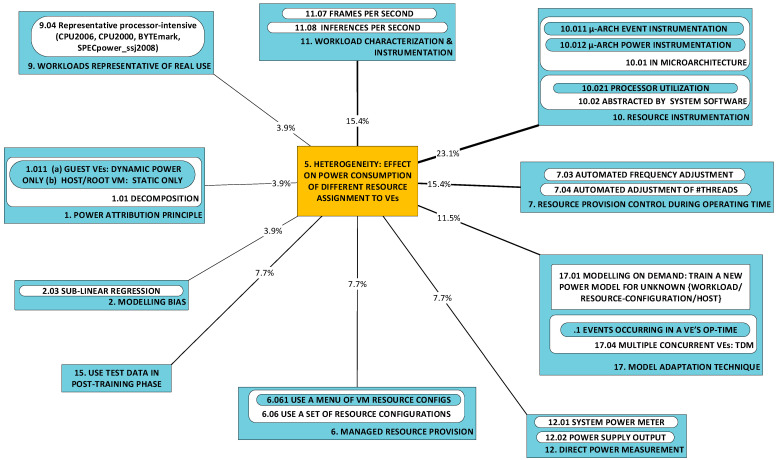
Approaches to solving challenges in category P5; utility metric $U_{A_{k}}$ is shown in line thickness and as a percentage. RUs in the respective categories are the following: P5 [62,72,78]; A1 [62]; A2 [62]; A6 [72]; A7 [78]; A9 [62]; A10 [62,78]; A11 [78]; A12 [62,72]; A15 [62]; A17 [62,78].

**Figure 14 sensors-23-00255-f014:**
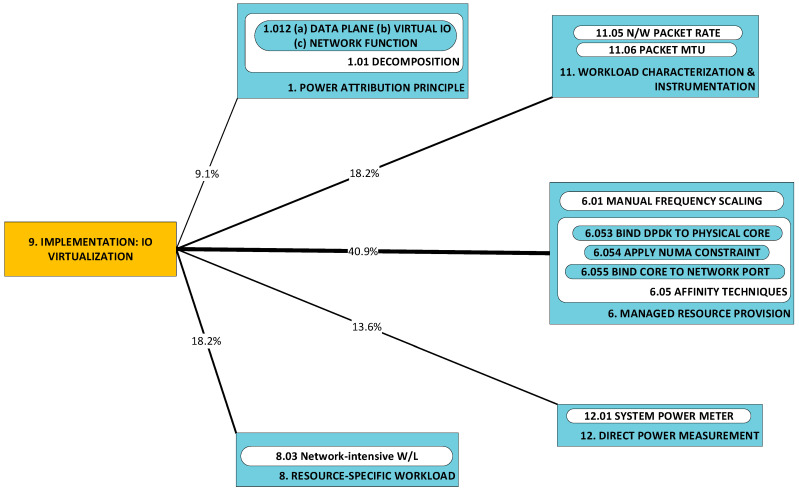
Approaches to solving challenges in category P9; utility metric $U_{A_{k}}$ is shown in line thickness and as a percentage. RUs in the respective categories are the following: P9 [51,53]; A1 [53]; A6 [51]; A8 [51,53]; A11 [53]; A12 [53].

**Figure 15 sensors-23-00255-f015:**
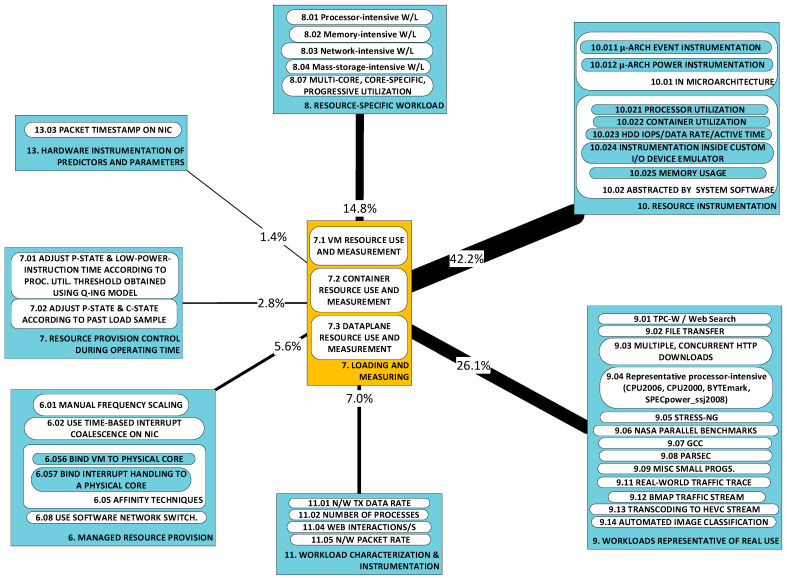
Approaches to solving challenges in category P7; utility metric $U_{A_{k}}$ is shown in line thickness and as a percentage. RUs in the respective categories are the following: P7 [56,57,58,59,61,62,65,68,69,71,72,75,76,77,78,79,81,82,83,84,85,87]; A6 [56,68,81,82,83]; A8 [56,59,61,71,76,77,84,85]; A9 [56,57,58,62,65,68,69,72,78,79,81,82,83,84,87]; A10 [56,57,58,59,61,62,69,72,75,76,77,78,79,81,82,83,84,85,87]; A11 [65,68,71,83,84,87]; A13 [87].

**Figure 16 sensors-23-00255-f016:**
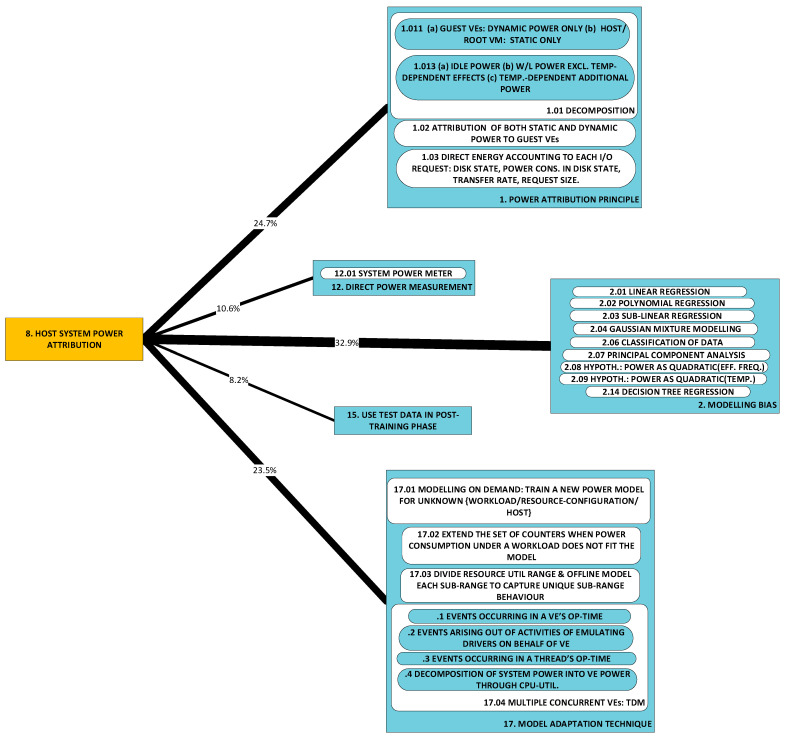
Approaches to solving challenges in category P8; utility metric $U_{A_{k}}$ is shown in line thickness and as a percentage. RUs in the respective categories are the following: P8 [57,58,59,61,62,69,72,75,76,77,79,81,82,84,86]; A1 [57,58,59,61,62,69,72,75,76,77,79,81,82,84,86]; A2 [57,58,59,61,62,72,75,76,77,79,81,84]; A12 [57,58,59,61,72,76,84]; A15 [57,62,77]; A17 [57,58,59,61,62,69,72,75,76,79,82,84,86].

**Figure 17 sensors-23-00255-f017:**
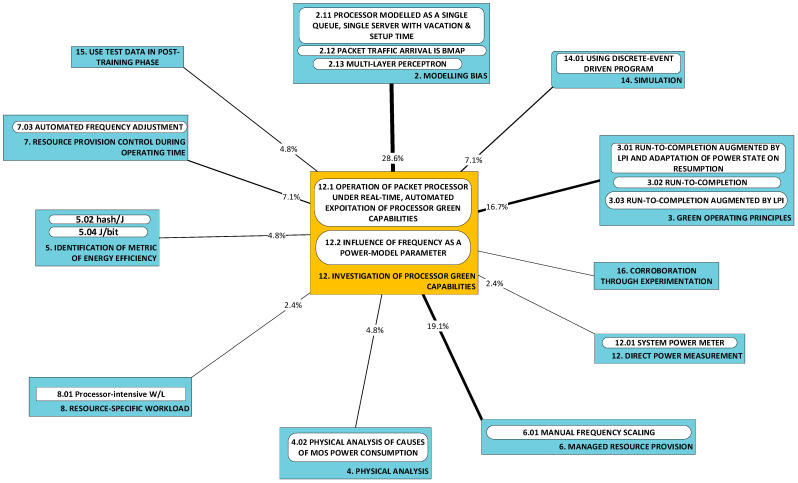
Approaches to solving challenges in category P12; utility metric $U_{A_{k}}$ is shown in line thickness and as a percentage. RUs in the respective categories are the following: P12: [50,60,67,81,82,83,85,87,94]; A2 [83,85,87,94]; A3 [83,87,94]; A4 [60]; A5 [50,85]; A6 [50,60,67,81,82]; A7 [85]; A8 [50]; A12 [50]; A14 [87]; A15 [85]; A16 [83].

**Figure 18 sensors-23-00255-f018:**
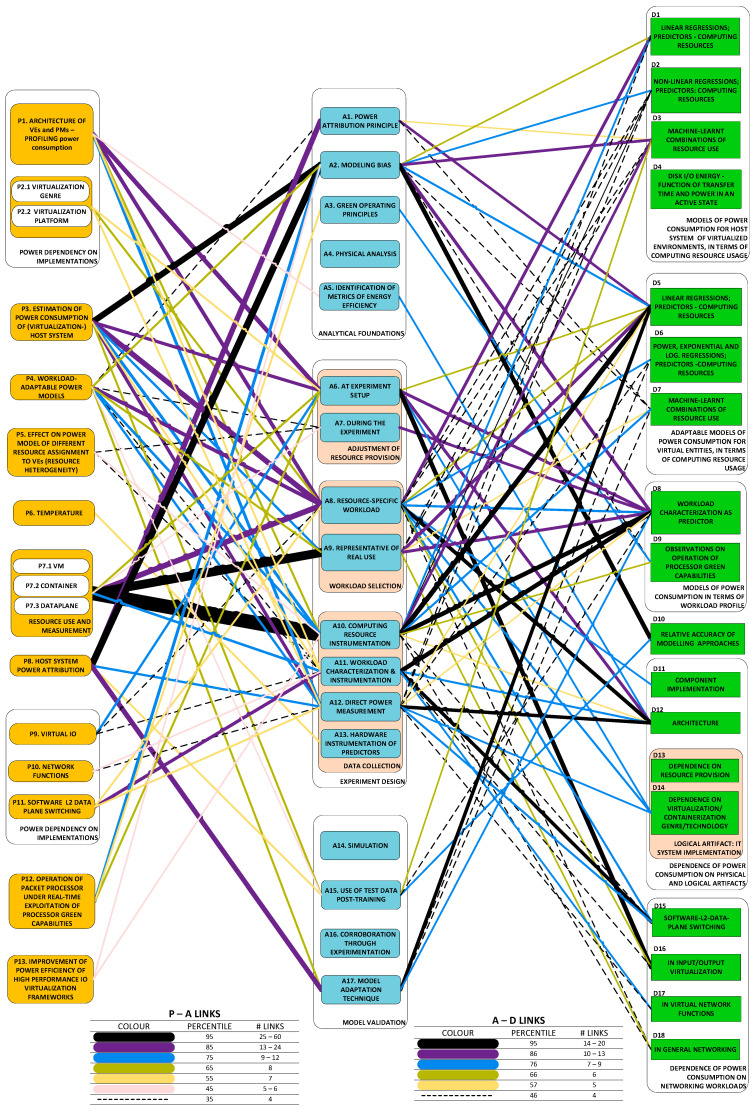
A directed acyclic graph showing the distribution of research into power measurement and power consumption models in virtualized networking and computing environments. D1 [57,58,59,69,75]; D2 [57,60,61,76,81]; D3 [62,77,79]; D5 [58,59,69,72,75,81,82,84,86]; D6 [61,66,76]; D7 [62,79]; D8 [63,64,65,66,67,68,71,78,83,85,87,94]; D9 [87]; D10 [60,62,63,77,79]; D11 [49,69,76]; D12 [49,50,67,68,76]; D13 [73]; D14 [55,56]; D16 [51,53]; D17 [48,53,85]; D18 [56,65].

**Figure 19 sensors-23-00255-f019:**
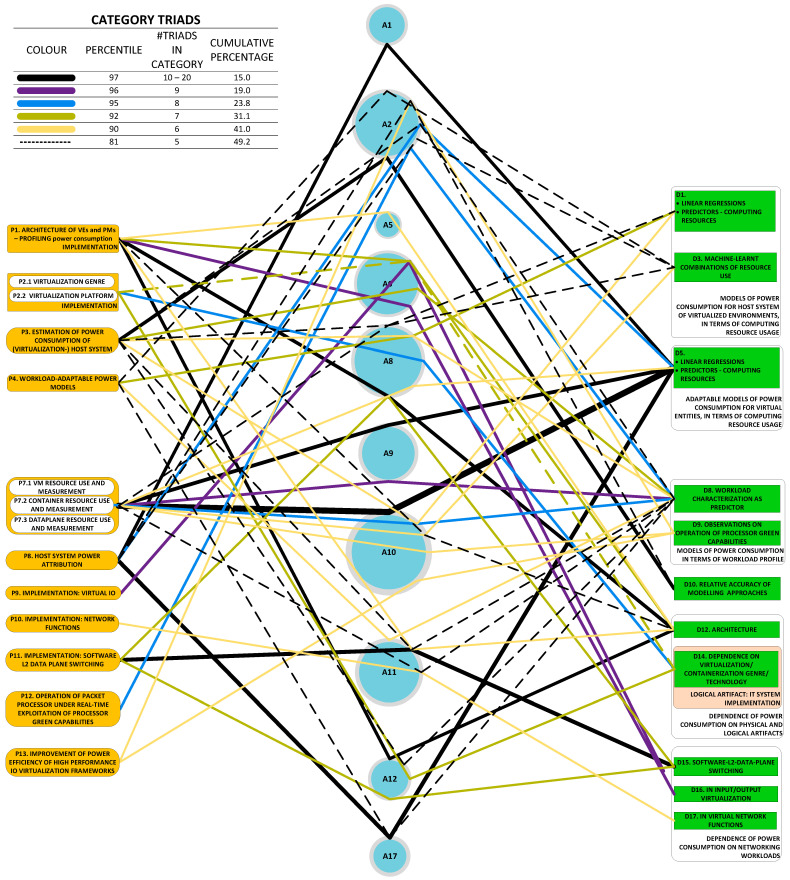
Triads. D1 [57,58,59,69,75]; D3 [62,77,79]; D5 [58,59,69,72,75,81,82,84,86]; D8 [63,64,65,66,67,68,71,78,83,85,87,94]; D9 [87]; D10 [60,62,63,77,79]; D11 [49,69,76]; D12 [49,50,67,68,76]; D14 [55,56]; D15 [48,53]; D16 [51,53]; D17 [48,53,85].

**Figure 20 sensors-23-00255-f020:**
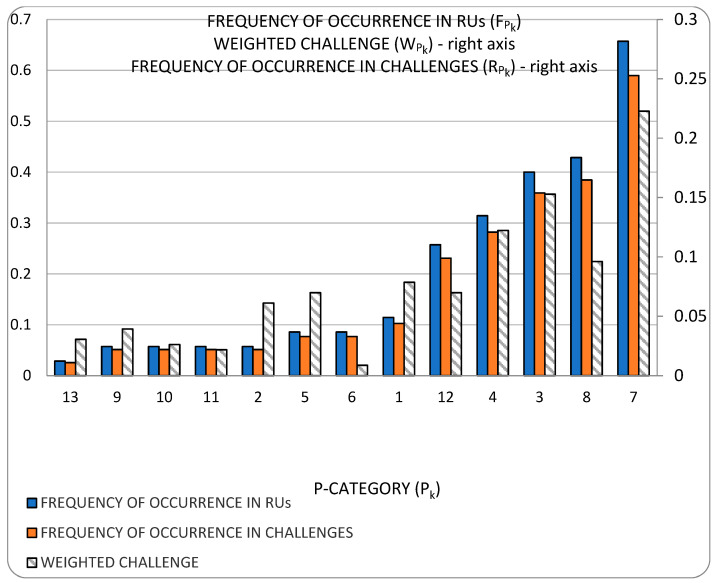
Frequency, Research Interest and weighted Challenge bar chart.

**Figure 21 sensors-23-00255-f021:**
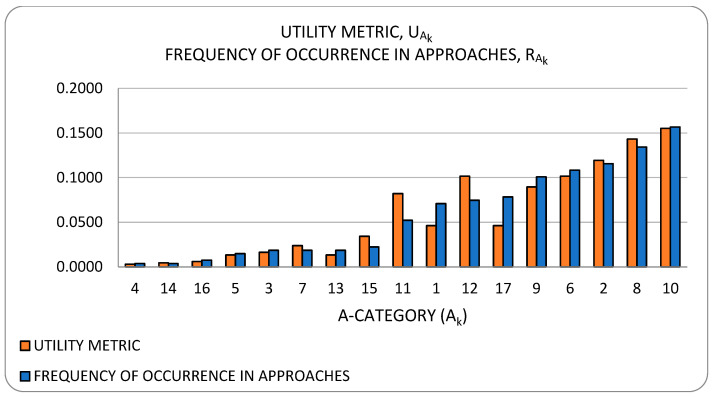
Approach metrics.

**Figure 22 sensors-23-00255-f022:**
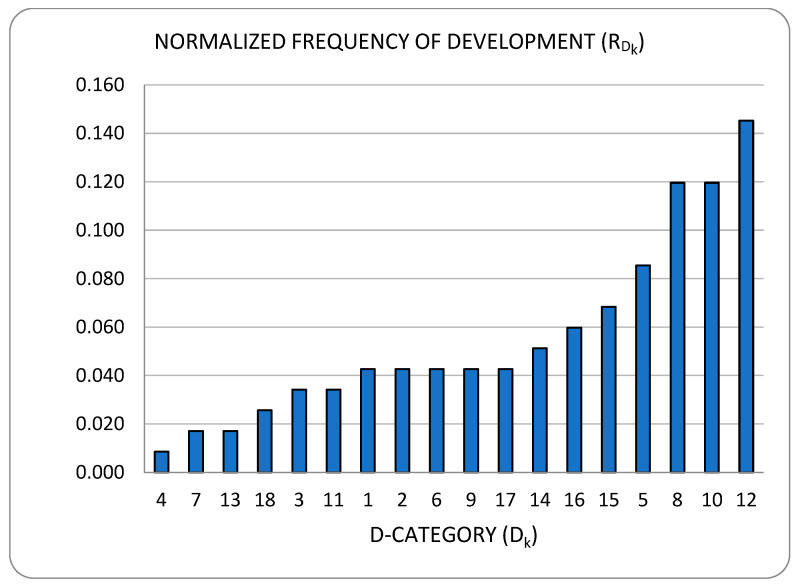
Normalized frequency of Developments.

**Figure 23 sensors-23-00255-f023:**
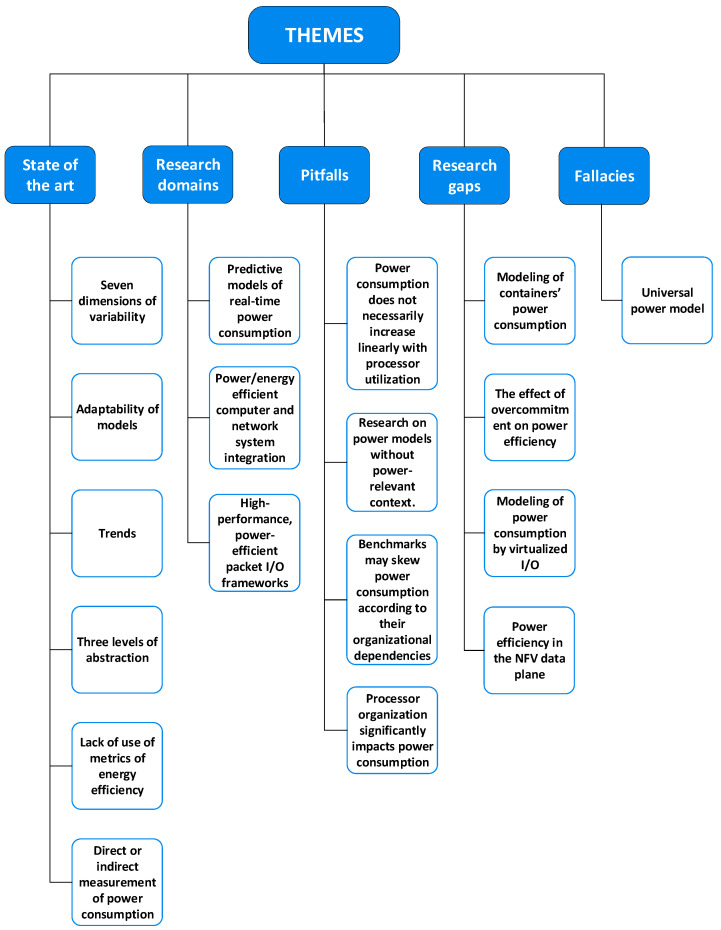
A graphical overview of the thematic analysis of the research space.

**Table 1 sensors-23-00255-t001:** Compound Annual Growth Rate (CAGR) reported in Cisco’s VNI over four consecutive years.

Period	Fixed Internet Traffic	Managed IP Traffic	Mobile Data
2014–2019 [6]	23	13	57
2015–2020 [7]	21	11	53
2016–2021 [8]	23	13	46
2017–2022 [9]	26	11	46

**Table 2 sensors-23-00255-t002:** Challenge category codes and representative challenge questions.

CAT #	CODE	PROBLEM
P1	Architecture of VEs and PMs—profiling power consumption	What is the impact of specific major architectural features of computer hardware on the power consumption of VEs?
P2	Virtualization genre, platform	Is it possible to meaningfully rationalize the behavior of VE power consumption across different implementations of systems for virtualization?
P3	Estimation of power consumption of virtualization-host systems	Can host system power consumption be predicted on the basis of VE activity?
P4	Workload-adaptable power models	Can workload-adaptable power models be developed?
P5	Resource heterogeneity	How do a VE’s power consumption and power model vary with resource configuration (heterogeneous VEs)?
P6	Influence of temperature on power model	How does temperature of operation affect VEs’ power models?
P7	Resource use and measurement of VEs	How can load be targeted at specific VE resources? How can the actual resource consumption be measured?
P8	Host system power attribution	How can the (measured) power consumption of a host be attributed to the hosted VEs?
P9	IO Virtualization	What is the impact of specific major implementations of IO virtualization on the power consumption of VEs?
P10	Network functions	Which particular implementation of a network function is most power or energy efficient?
P11	Software L2 data plane switching	What is the impact of specific major implementations of software layer 2 data plane switching on the power consumption of VEs?
P12	Investigation of processor green capabilities	How can we model operation under real-time exploitation of processor green capabilities?
P13	Improvement of power efficiency of high-performance IO virtualization	How can we improve the power efficiency of high-performance packet IO frameworks?

**Table 3 sensors-23-00255-t003:** Approach category codes.

CAT #	CODE
A1	Power attribution principle
A2	Modeling bias
A3	Green operating principles
A4	Physical analysis
A5	Identification and use of metrics of energy efficiency
A6	Managed resource provision (setup time)
A7	Controlled resource provision (operation time)
A8	Resource-specific workloads
A9	Workloads representative of real use
A10	Computing resource instrumentation
A11	Workload characterization and instrumentation
A12	Direct power measurement
A13	Hardware instrumentation of predictors
A14	Simulation
A15	Use of test data in post-training phase
A16	Corroboration through experimentation
A17	Model adaptation technique

**Table 4 sensors-23-00255-t004:** Development category codes.

CAT #	CODE
D1	Host models -> linear regressions: predictors = computing resources
D2	Host models -> non-linear regressions: predictors = computing resources
D3	Host models -> machine-learnt: inputs = computing resources
D4	Host models -> mass storage energy consumption
D5	Adaptable VE models -> linear regressions: predictors = computing resources
D6	Adaptable VE models -> power, exponential and log regressions: predictors = computing resources
D7	Adaptable VE models -> machine-learnt: inputs = computing resources
D8	Host/VE models of power consumption -> predictors = workload characteristics
D9	Host/VE models of power consumption- > observations on operation of processor green capabilities
D10	Relative accuracy of modeling approaches
D11	Power’s dependencies- > physical and logical artifacts -> component implementation
D12	Power’s dependencies -> physical and logical artifacts -> architecture
D13	Power’s dependencies -> physical and logical artifacts -> IT system implementation -> dependence on resource provision
D14	Power’s dependencies -> physical and logical artifacts -> IT system implementation -> dependence on virtualization/containerization genre and technology
D15	Power’s dependencies -> networking workloads -> software L2 data plane switching
D16	Power’s dependencies- > networking workloads -> in I/O virtualization
D17	Power’s dependencies- > networking workloads -> in VNFs
D18	Power’s dependencies -> networking workloads -> in general networking

**Table 5 sensors-23-00255-t005:** Seven dimensions of variability.

Dimension	Problem Category
1. Workload type	P4, P7
2. Virtualization agent	P2, P9, P11, P13
3. Host (resources and architecture)	P1, P3
4. Temperature	P6
5. Power attribution	P8
6. Co-hosted, concurrent VEs	P8, P5
7. Frequency	P12, P13

## Data Availability

**The data presented in this study are available in**https://github.com/humaira-salam/PowerMeasurementAndModelingRawData.

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
