# Peer review of "Dynamics of Research into Modeling the Power Consumption of Virtual Entities Used in the Telco Cloud"

_sensors, 2022, doi:10.3390/s23010255_

Round 1
Reviewer 1 Report
This work provides dense review on the topic of power consumption modeling for telco networks. In general, the paper is well organized and contains enough contribution as a review paper. I have few comments for further improving the quality of this work.
1. Please make sure every table and figure are cited and discussed in the text part.
2. An abbreviation table is required to improve the readability.
3. Green communication is a popular topic in recent decades. More discussion should be made on the potential application of this review work on the field of green communications. This will also attracts readers from ComSoc. The following paper may be helpful for citation:
Full-Duplex Versus Half-Duplex Amplify-and-Forward Relaying: Which is More Energy Efficient in 60-GHz Dual-Hop Indoor Wireless Systems? IEEE JSAC
Author Response
All noted, with thanks. Action will be taken in accordance with suggestions.
OBSERVATION 1. :Please make sure every table and figure are cited and discussed in the text part.
REPLY TO OBS 1: It seems that every table and figure is cited and discussed in the text part.
OBSERVATION 2: An abbreviation table is required to improve the readability
REPLY TO OBS.2: Thank you for pointing this out. This has been added as "Appendix B"
OBSERVATION 3: Green communication is a popular topic in recent decades. More discussion ... may be helpful for citation.
REPLY TO OBS 3: Thank you for pointing this out. We have included a PAD analysis of a popular area, i.e., use of renewable energy sources to power radio access network base stations. This has been included as an appendix since our contributions lie in real-time predictive models of power consumption in virtual entities (whether machines or containers), with emphasis on the use of such VEs in computer systems hardware and software that are pertinent to the telco cloud. Therefore, we respectfully suggest limited coverage (one area), in a body of text (an appendix) separate from the principal one (the main body), using PAD as a means of continuity between the appendix and the main body
Reviewer 2 Report
The authors present an extensive review on power consumption modelling in virtualized network environments, wherein it is highlighted that the measurements presented in literature indicate that power consumption models are non-linear, contrary to linear models that have been widely applied in literature. Nonetheless, the modelling of power consumption in virtualized networks in not uncomplicated, and developing general models for all network scenarios may not be realistic. Models may therefore apply to specific environments. The following highlight the main points:
* The underlying topic is timely and relevant to the research community.
* The manuscript requires minor revisions to fix the typos and to improve the writing style.
* The length of the manuscript should be reviewed in line with the submission and publication guidelines by the journal.
Author Response
All noted, with thanks. Action will be taken in accordance with suggestions.
OBSERVATION 1: The underlying topic is timely and relevant to the research community.
REPLY TO OBS 1: Noted, with thanks.
OBSERVATION 2: The manuscript requires minor revision to fix the typos and to improve the writing style.
REPLY TO OBS 1: Noted, with thanks. We have homogenized orthography to US English, and written the word "modeliing" uniformly throughout, as this was sometimes spelt with a single letter "l" and sometimes with a double letter "l". If there are further corrections that we have missed, we would be grateful for further indications, so as to correct them.
OBSERVATION 3: The length of the manuscript should be reviewed in line with the submission and publication guidelines by the journal.
REPLY TO OBS 3: Noted, with thanks. We understand that the journal does not limit the number of pages. We would be grateful for any suggestions, if relevant, on whether some material may be considered redundant.